# A genome-wide arrayed CRISPR screen identifies PLSCR1 as an intrinsic barrier to SARS-CoV-2 entry that recent virus variants have evolved to resist

Jérémie Le Pen[1][�][*], Gabrielle Paniccia[1][�][¤a], Volker Kinast[2,3][‡], Marcela Moncada-Velez[4][‡], Alison W. Ashbrook[1][‡], Michael Bauer[1][‡], H.-Heinrich Hoffmann[1][‡], Ana Pinharanda[5][‡], Inna Ricardo-Lax[1][‡], Ansgar F. Stenzel[1][¤b][‡], Edwin A. Rosado-Olivieri[6][¤c][‡], Kenneth H. Dinnon, III[1], William C. Doyle[1][¤d], Catherine A. Freije[1], Seon-Hui Hong[1], Danyel Lee[4,7,8], Tyler Lewy[1], Joseph M. Luna[1][¤e], Avery Peace[1], Carltin Schmidt[4][¤f], William M. Schneider[1], Roni Winkler[1], Elaine Z. Yip[1], Chloe Larson[9], Timothy McGinn[9], Miriam-Rose Menezes[9], Lavoisier Ramos-Espiritu[9], Priyam Banerjee[10], John T. Poirier[11], Francisco J. Sànchez-Rivera[12,13], Aurélie Cobat[4,7,8], Qian Zhang[4,7,8], Jean-Laurent Casanova[4,7,8,14,15], Thomas S. Carroll[16], J. Fraser Glickman[9], Eleftherios Michailidis[1][¤g][‡], Brandon Razooky[1][¤h][‡], Margaret R. MacDonald[1][‡], Charles M. Rice[1][‡][*]

1 Laboratory of Virology and Infectious Disease, The Rockefeller University, New York, New York, United States of America, 2 Department of Medical Microbiology and Virology, Carl von Ossietzky University Oldenburg, Oldenburg, Germany, 3 Department for Molecular and Medical Virology, Faculty of Medicine, Ruhr University Bochum, Bochum, Germany, 4 St Giles Laboratory of Human Genetics of Infectious Diseases, Rockefeller Branch, The Rockefeller University, New York, New York, United States of America, 5 Department of Biological Sciences, Columbia University, New York, New York, United States of America, 6 Laboratory of Synthetic Embryology, The Rockefeller University, New York, New York, United States of America, 7 Laboratory of Human Genetics of Infectious Diseases, Necker Branch, INSERM U1163, Paris, France, 8 Paris Cité University, Imagine Institute, Paris, France, 9 Fisher Drug Discovery Resource Center, The Rockefeller University, New York, New York, United States of America, 10 Bio-Imaging Resource Center, The Rockefeller University, New York, New York, United States of America, 11 Laura and Isaac Perlmutter Cancer Center, New York University Grossman School of Medicine, NYU Langone Health, New York, New York, United States of America, 12 Department of Biology, Massachusetts Institute of Technology, Cambridge, Massachusetts, United States of America, 13 David H. Koch Institute for Integrative Cancer Research, Massachusetts Institute of Technology, Cambridge, Massachusetts, United States of America, 14 Department of Pediatrics, Necker Hospital for Sick Children, Paris, France, 15 Howard Hughes Medical Institute, New York, New York, United States of America, 15 Bioinformatics Resource Center, The Rockefeller University, New York, New York, United States of America

☉ These authors contributed equally to this work.
¤a Current address: Elya Laboratory, Department of Molecular and Cellular Biology, Harvard University, Cambridge, Massachusetts, United States of America
¤b Current address: Department of Infectious Diseases, Molecular Virology, Heidelberg University, Heidelberg, Germany
¤c Current address: Department of Molecular Pathobiology, New York University, New York, United States of America
¤d Current address: Laboratory of Host-Pathogen Biology, The Rockefeller University, New York, New York, United States of America
¤e Current address: Department of Biochemistry and Center for RNA Science and Therapeutics, Case Western Reserve University, Cleveland, Ohio, United States of America
¤f Current address: Medical Faculty, Heinrich Heine University Dusseldorf, Germany
¤g Current address: Laboratory of Biochemical Pharmacology, Department of Pediatrics, Emory University School of Medicine, Atlanta, Georgia, United States of America
¤h Current address: National Resilience, Inc., La Jolla, California, United States of America
‡ VK, MM-V, AWA, MB, H-HH, AP, IR-L, AFS, and EAR-O also contributed equally to this work. EM, BR, MRM, and CMR also contributed equally to this work.
* jlepen@rockefeller.edu (JLP); ricec@rockefeller.edu (CMR)

**Data Availability Statement:** mRNA-seq data is available on the Sequence Read Archive (SRA), hosted by the National Center for Biotechnology

Information (NCBI), under BioProject PRJNA1138251 and BioSamples SAMN42689767-SAMN42689770 corresponding to S2 and S3 Tables, BioSamples SAMN42689771-SAMN42689774 corresponding to S4 and S5 Tables, BioSamples SAMN42689775-SAMN42689776 corresponding to S14 and S15 Tables, BioSamples SAMN42689777-SAMN42689778 corresponding to S16 and S17 Tables. Code and Supporting Information are available on Dryad (DOI: 10.5061/dryad. 6q573n65k).

**Funding:** The Laboratory of Virology and Infectious Disease was supported by the National Institutes of Health (P01AI138398-S1, 2U19AI111825, R01AI091707-10S1, and R01AI161444 to CMR); a George Mason University Fast Grant to CMR; the G. Harold and Leila Y. Mathers Charitable Foundation to CMR; the Meyer Foundation to CMR; the Pilot Project Robertson Therapeutic Development Fund at The Rockefeller University to CMR; and the Bawd Foundation to CMR. The Laboratory of Human Genetics of Infectious Diseases was supported by the Howard Hughes Medical Institute to JLC, the St. Giles Foundation to JLC, the National Institutes of Health (R01AI163029 and UL1TR001866 to JLC), the French National Research Agency (ANR) under the "Investments for the Future" program (ANR-10-IAHU-01 to JLC), the Integrative Biology of Emerging Infectious Diseases Laboratory of Excellence (ANR-10-LABX-62-IBEID to JLC), the French Foundation for Medical Research (FRM) (EQU201903007798 to JLC), ANR GENVIR (ANR-20-CE93-003) to JLC, ANR AAILC (ANR-21-LIBA-0002) to QZ, ANR GENFLU (ANR-22-CE92-0004) to QZ, the ANR-RHU COVIFERON Program (ANR-21-RHUS-08) to JLC, the French Foundation for Medical Research (FRM) (EQU201903007798) to JLC, the European Union's Horizon 2020 research and innovation program under grant agreement No. 824110 (EASI-genomics) to JLC, the HORIZON-HLTH-2021-DISEASE-04 program under grant agreement 101057100 (UNDINE) to JLC, the Square Foundation to JLC, William E. Ford, General Atlantic's Chairman and Chief Executive Officer, Gabriel Caillaux, General Atlantic's Co-President, Managing Director and Head of Business at EMEA, and the General Atlantic Foundation to JLC, the French Ministry of Higher Education, Research, and Innovation (MESRI-COVID-19) to JLC, and REACTing-INSERM to JLC. JLP was supported by the Francois Wallace Monahan Postdoctoral Fellowship at The Rockefeller University and the European Molecular Biology Organization Long-Term Fellowship (ALTF 380-2018 to JLP). GP was supported by the James

# Abstract

Interferons (IFNs) play a crucial role in the regulation and evolution of host–virus interactions. Here, we conducted a genome-wide arrayed CRISPR knockout screen in the presence and absence of IFN to identify human genes that influence Severe Acute Respiratory Syndrome Coronavirus 2 (SARS-CoV-2) infection. We then performed an integrated analysis of genes interacting with SARS-CoV-2, drawing from a selection of 67 large-scale studies, including our own. We identified 28 genes of high relevance in both human genetic studies of Coronavirus Disease 2019 (COVID-19) patients and functional genetic screens in cell culture, with many related to the IFN pathway. Among these was the IFN-stimulated gene *PLSCR1*. PLSCR1 did not require IFN induction to restrict SARS-CoV-2 and did not contribute to IFN signaling. Instead, PLSCR1 specifically restricted spike-mediated SARS-CoV-2 entry. The PLSCR1-mediated restriction was alleviated by TMPRSS2 overexpression, suggesting that PLSCR1 primarily restricts the endocytic entry route. In addition, recent SARS-CoV-2 variants have adapted to circumvent the PLSCR1 barrier via currently undetermined mechanisms. Finally, we investigate the functional effects of PLSCR1 variants present in humans and discuss an association between PLSCR1 and severe COVID-19 reported recently.

## Introduction

Viruses maintain a complex relationship with their host cells, co-opting host factors for their replication while being targeted by cellular defense mechanisms. Such cellular defenses include the interferon (IFN) pathway, where the infected cell senses foreign molecules and secretes IFN to trigger an antiviral state in neighboring cells [1].

Approximately 1% to 5% of critical Coronavirus Disease 2019 (COVID-19) patients have mutations that compromise the production of or response to type I IFNs, while an additional 15% possess autoantibodies that neutralize type I IFNs [2–8]. This highlights the essential role of type I IFN in the defense against the Severe Acute Respiratory Syndrome Coronavirus 2 (SARS-CoV-2) virus that caused the COVID-19 pandemic [9,10]. Consequently, investigating IFN-stimulated genes (ISGs) is crucial to our understanding of the remarkable antiviral systems that evolved in nature and could enhance our preparedness for future pandemics.

Several recent studies have identified ISGs restricting SARS-CoV-2. Most of these studies involved gain-of-function genetic screens, overexpressing individual ISGs. The factors bone marrow stromal cell antigen 2 (BST2), cholesterol 25-hydroxylase (CH25H), lymphocyte antigen 6 family member E (LY6E), 2′-5′-oligoadenylate synthetase 1 (OAS1), and receptor transporter protein 4 (RTP4) were notably identified as SARS-CoV-2 antivirals in these studies [10–15]. One advantage of the gain-of-function approach is that it circumvents potential genetic redundancies between ISGs [16,17]. However, this approach is biased towards ISGs that act autonomously when overexpressed and does not mimic the cellular context of the IFN response, where hundreds of genes and gene products are differentially regulated to establish an antiviral state. To counter this limitation, 2 recent publications examined the effects of ISG loss of function in IFN-treated cells. They conducted pooled CRISPR knockout (KO) screens in cells pretreated with IFN before SARS-CoV-2 infection [18,19]. By sorting for cells with

H. Gilliam Fellowship for Advanced Study from the Howard Hughes Medical Institute and the Graduate Research Fellowship Program from the National Science Foundation (FAIN 1946429 to GP). VK was supported by a travel grant of the Boehringer Ingelheim Fonds (BIF) and a scholarship of the German Liver Foundation. MB was supported by a Swiss National Science Foundation fellowship (P500PB_203007 to MB). DL was supported by the European Society for Immunodeficiencies bridge grant and a fellowship from the FRM. The funders had no role in study design, data collection and analysis, decision to publish, or preparation of the manuscript.

**Competing interests:** The authors have declared that no competing interests exist.

**Abbreviations:** ACE2, angiotensin converting enzyme 2; AP-MS, affinity purification-mass spectrometry; BST2, bone marrow stromal cell antigen 2; CHIKV, chikungunya virus; CH25H, cholesterol 25-hydroxylase; COVID-19, Coronavirus Disease 2019; CTSL, cathepsin L; DMEM, Dulbecco's Modified Eagle Medium; FBS, fetal bovine serum; FFU, focus-forming unit; GSEA, gene set enrichment analysis; GWAS, genome-wide association study; HIV-1, human immunodeficiency virus type 1; hPIV, human parainfluenza virus; HSV-1, herpes simplex virus 1; IAV, influenza A virus; IFN, interferon; IRF9, IFN regulatory factor 9; ISG, IFN-stimulated gene; JAK1, Janus kinase 1; KO, knockout; LY6E, lymphocyte antigen 6 family member E; MAF, minor allele frequency; NCOA7, nuclear receptor coactivator 7; NEAA, nonessential amino acid; NLS, nuclear localization signal; OAS1, 2′-5′-oligoadenylate synthetase 1; PBS, phosphate buffered saline; PLSCR1, phospholipid scramblase 1; RTP4, receptor transporter protein 4; SARS-CoV-2, Severe Acute Respiratory Syndrome Coronavirus 2; SINV, Sindbis virus; SOCS1, suppressor of cytokine signaling 1; SREBF2, sterol regulatory element binding transcription factor 2; STAT2, signal transducer and activator of transcription 2; TMPRSS2, transmembrane serine protease 2; TYK2, tyrosine kinase 2; USP18, ubiquitin-specific peptidase 18; VEEV, Venezuelan equine encephalitis virus; VSV, vesicular stomatitis virus; WB, western blot; WT, wild type; Y2H, yeast two-hybrid.

high SARS-CoV-2 viral load, they identified SARS-CoV-2 restriction factors such as death domain associated protein (DAXX).

Here, we conducted a human whole-genome arrayed CRISPR KO screen to identify genes that influence SARS-CoV-2 infection in cells with or without pretreatment with a low dose of IFN. The arrayed approach, though logistically challenging, has advantages over the pooled format in capturing both proviral and antiviral genes, genes affecting virus egress, and those coding for secreted products that exert their impact on neighboring cells. It reliably captures genotype–phenotype correlations while also unveiling the effects of single gene perturbation on cell growth and death [20]. Additionally, the shorter culture time and lack of competition among cells with different gene KO in the arrayed screen allow the inclusion of genes that would be depleted and deemed essential in a pooled format [21]. The arrayed format thus enables the identification of crucial cellular functions that may be co-opted by the virus or are vital for the cell's defense against infection.

We then compiled a comprehensive list of genes interacting with SARS-CoV-2, incorporating findings from our own screen as well as existing literature. This meta-analysis revealed several host genes of interest, both previously described and novel. Notably, the ISG product phospholipid scramblase 1 (PLSCR1) emerged as a prominent antiviral factor. PLSCR1 is involved in several biological processes [22], including regulating the movement of phospholipids between the 2 leaflets of a cell membrane (lipid scrambling) [23] and IFN signaling in the context of virus infection [24].

Follow-up experiments revealed that PLSCR1 is a cell intrinsic factor that restricts spike-mediated SARS-CoV-2 entry, independently of the IFN pathway, via currently undetermined mechanisms. Our genetic screen data and meta-analysis provide a valuable resource to broaden our understanding of coronavirus infection and innate immunity. Furthermore, we extend the recent characterization of PLSCR1 as an antiviral against SARS-CoV-2 impacting COVID-19 outcomes (**S1 Fig**) [19,25,26].

# Results

## A genome-wide arrayed CRISPR KO screen identifies known and novel factors influencing SARS-CoV-2 infection

While the liver is not the primary target organ of SARS-CoV-2 infection, human hepatocellular carcinoma Huh-7.5 cells naturally express SARS-CoV-2 dependency factors, including the receptor angiotensin converting enzyme 2 (ACE2), and proved unexpectedly useful in SARS-CoV-2 research [19,27–34]. Huh-7.5 cells are defective in virus sensing and do not commonly produce IFN during infection [35]. We confirmed that Huh-7.5 cells do not induce ISG expression during SARS-CoV-2 infection (**Fig 1A** and **S2** and **S3 Tables**). This is likely due to a defect upstream of IFN production, as these cells did induce ISG expression when treated with recombinant IFN (**Fig 1B** and **S4** and **S5 Tables**). Thus, Huh-7.5 cells are a convenient model for studying controlled IFN responses during viral infection. Furthermore, IFN treatment restricted SARS-CoV-2 (**Fig 1C**), indicating that some ISGs are effective in limiting SARS-CoV-2 infection in Huh-7.5 cells. This allows us to study the functional landscape of SARS-CoV-2 restriction, examining both intrinsic factors and those induced in response to IFN signaling. Similar to A549-ACE2 cells [11,36], Huh-7.5 cells do not express transmembrane serine protease 2 (TMPRSS2) (**S1B Fig**). As a result, SARS-CoV-2 entry is restricted to the endocytic route [37,38].

Using these cells, we conducted a whole-genome arrayed CRISPR KO screen designed to identify both SARS-CoV-2 proviral and antiviral genes, whose KO reduces or enhances SARS-CoV-2 infection, respectively. In particular, we aimed to identify factors involved in the IFN response, from IFN sensing to ISG induction, including effector ISGs that directly

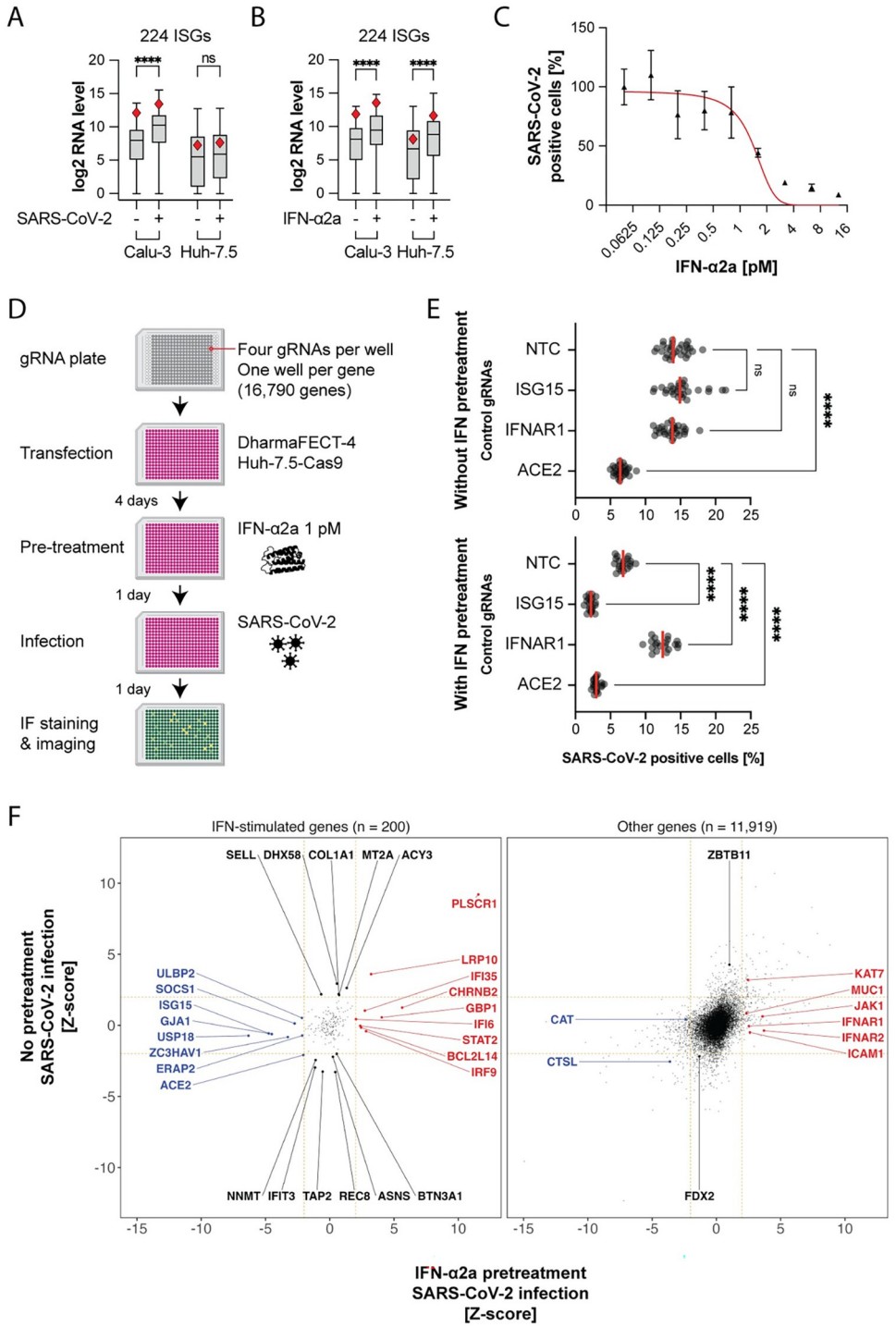

**Fig 1. An unbiased arrayed CRISPR KO screen identifies IFN-dependent and IFN-independent genes influencing SARS-CoV-2 infection.** (**A**) mRNA-seq comparison between Huh-7.5 and Calu-3 cells, focusing on a subset of 224 ISGs, in response to 24-hour SARS-CoV-2 infection MOI 0.03. Red diamond, PLSCR1 RNA level. Viral RNA levels were comparable in both cell lines (not shown). ****, $p \leq 0.0001$; two-tailed $t$ test. (**B**) Cells were treated with 0.5 nM IFN-α2a for 24 hours. mRNA-seq analysis as in (**A**). C. Huh-7.5-Cas9 cells were pretreated with different amounts of IFN-α2a and then infected with SARS-CoV-2 for 24 hours followed by IF staining for SARS-CoV-2 N protein; $n = 6$ separate wells infected on the same day; error bars represent SEM; ****, $p \leq 0.0001$; two-tailed $t$ test. (**D**) Diagram of the arrayed CRISPR KO screen method. (**E**) Median SARS-CoV-2-positive cell percentages determined by IF staining for the indicated control genes. The control genes were distributed across 28 separate wells (without IFN pretreatment)

or across 20 separate wells (with IFN pretreatment) for each screened library plate. (**F**) The virus level (percentage of infected cells normalized and z-score calculated) is plotted for 24 hours 0 pM (*y* axis) or 1 pM (*x* axis) IFN-α2a pretreatment followed by 24-hour infection ($n \geq 5$). The genes were categorized as ISG or other based on mRNA-seq of IFN-α2a-treated cells as in (**B**). ISGs were defined by a fold change $\geq 2$ and padj $\leq 0.05$ in the IFN-treatment versus untreated pairwise comparison. The data underlying this Figure can be found in **S1 Table**. IF, immunofluorescence; IFN, interferon; ISG, IFN-stimulated gene; KO, knockout; MOI, multiplicity of infection; NTC, non-targeting control; SARS-CoV-2, Severe Acute Respiratory Syndrome Coronavirus 2.

influence the virus life cycle. With this arrayed approach performed in 384-well plates, the cells in each well received a pool of 4 gRNAs targeting a single host gene. Each well was then either treated with 1 pM IFN-α2a or left untreated before being infected with SARS-CoV-2, followed by SARS-CoV-2 nucleoprotein (N) immunofluorescence staining and high content microscopy (**Fig 1D**). This low dose of IFN was chosen to mimic a cellular environment where IFN triggers an antiviral state before infection. We anticipated that a saturating amount of IFN would lead to high ISG transcription and functional redundancy between effectors, biasing hits towards factors in IFN signaling. In contrast, a low dose of IFN, around the IC50, might enable identification of the specific roles of individual effector ISGs.

We conducted the functional assays 4 days after KO to allow sufficient time for full protein depletion, considering both cell division and protein half-life (**S2E Fig**).

We included control gRNAs with known proviral and antiviral effects in our screening plates. As expected, the SARS-CoV-2 receptor ACE2 [9] behaved as a proviral gene with and without IFN pretreatment. The IFN-alpha/beta receptor alpha chain 1 (IFNAR1) [39–41] was essential for the IFN-mediated restriction of SARS-CoV-2. The ISG15 ubiquitin-like modifier (ISG15) has been previously described as a negative regulator of the IFN response. ISG15 stabilizes ubiquitin-specific peptidase 18 (USP18) [42–44], which, in turn, binds to the IFN-alpha/beta receptor alpha chain 2 (IFNAR2) and blocks signal transduction [45,46]. Accordingly, ISG15 was proviral in IFN-treated cells (**Fig 1E**).

Of the 16,790 screened genes, we selected 16,178 genes where KO did not lead to changes in cellular fitness, as assessed by nuclei count ($-2 \leq$ z-score $\leq 2$) (**S2B Fig**). From these, we identified 12,119 genes expressed in 3 cell lines relevant for SARS-CoV-2 research (A549, Calu-3, and Huh-7.5 cells) and human lung cells, the primary target cell type in vivo, based on previously published data [47–49] (see S6 Table for a list of expressed genes). These genes were selected for downstream analysis. We then binned the genes into 2 groups for data visualization, depending on whether they were induced by IFN-α2a treatment in Huh-7.5 cells as determined by mRNA-seq (log 2 FC $\geq 2$ and padj $\leq 0.05$) (**S2C Fig** and **S7–S9 Tables**).

We classified 448 genes with a SARS-CoV-2 infection z-score $\geq 2$ as antiviral hits and 507 genes with a SARS-CoV-2 infection z-score $\leq -2$ as proviral hits (**S9 Table**).

Our screen found known and previously unidentified host factors influencing SARS-CoV-2 infection (**Fig 1F**). As expected, positive regulators of IFN signaling, such as IFNAR1,2 [39–41], IFN regulatory factor 9 (IRF9) [50,51], Janus kinase 1 (JAK1) [52], and signal transducer and activator of transcription 2 (STAT2) [53,54] were antiviral only in IFN-pretreated cells. Known negative regulators of IFN signaling, such as ISG15 [42–44], suppressor of cytokine signaling 1 (SOCS1) [55], and USP18 [45,46] were proviral only in IFN-pretreated cells.

The receptor ACE2 was required for infection [9]. In our mRNA-seq analysis, IFN treatment was found to significantly up-regulate ACE2 mRNA levels (**S2C Fig**). Prior studies indicate that IFN induces transcription of a truncated ACE2 isoform, rather than the full-length receptor for SARS-CoV-2 [56,57]. The lysosomal cysteine protease cathepsin L (CTSL), required for SARS-CoV-2 spike protein activation during endocytosis [58–60], was also a proviral hit in our screens (**Fig 1F**).

The screen data likely contains false negatives. For example, STAT1 and tyrosine kinase 2 (TYK2) [61,62] did not influence infection alongside other positive regulators of IFN signaling, which we attribute to the fact that some gRNAs in the library may have not efficiently directed Cas9 to cut at their respective target gene loci.

Collectively, the identification of known proviral and antiviral factors confirms the validity of our screening method.

We performed a gene set enrichment analysis (GSEA) to identify cellular pathways exhibiting proviral or antiviral properties in our screen. The full GSEA results, including the genes driving each pathway enrichment (so-called "leading edge"), can be found in **S10 Table**. Some top pathways ranked by adjusted *p*-value are summarized in **S2D Fig**. Notably, pathways associated with RNA pol II transcription and mRNA maturation, as well as pathways related to cellular respiration, exhibited antiviral activity independent of IFN (**Fig 2**). Surprisingly, RNA pol III transcription, in part driven by the genes RNA polymerase III subunit A (*POLR3A*) and RNA polymerase III subunit B (*POLR3B*), were critical to the antiviral response mediated by IFN. Conversely, factors involved in translation, such as eukaryotic translation initiation factor 3 subunits F and G (EIF3G and EIF3F), likely co-opted for producing viral proteins, were identified as proviral. Similarly, factors regulating cholesterol homeostasis, likely crucial for SARS-CoV-2 entry [14,63], were also identified as proviral. For instance, the gene sterol regulatory element binding transcription factor 2 (*SREBF2*) was one of the top proviral genes (**Fig 2**).

Our arrayed CRISPR KO screen results constitute a valuable resource for research on coronavirus infection specifically, as well as on innate immunity in general. These can be used to help characterize human genes influencing SARS-CoV-2 infection and the IFN response.

## The ISG PLSCR1 is associated with COVID-19 outcomes and exhibits antiviral effects in functional SARS-CoV-2 genetic screens

To provide a thorough perspective on human genes that impact SARS-CoV-2 infection and to place our arrayed CRISPR KO screen results within the context of existing research, we have compiled a table that includes findings from a selection of 67 large-scale "omic" studies related to SARS-CoV-2. This compilation encompasses this study and 25 other functional genetic screens for genes that influence SARS-CoV-2 infection [11–13,15,18,19,28,31,64–80], 24 human genetic studies that correlate certain alleles with severe COVID-19 outcomes [5,6,8,25,26,81–99], 10 publications detailing SARS-CoV-2 protein interactomes [100–109], 6 focusing on SARS-CoV-2 RNA interactomes [110–115], and one that examines proteins with altered phosphorylation states in SARS-CoV-2-infected cells [116] (**S11 and S12 Tables for the full and summary tables, respectively**). This table highlights the depth of research in publications addressing SARS-CoV-2 infection: Genes reported in several independent large-scale studies are more credible candidates for biological relevance (**S3 Fig**). As expected, genes associated with the IFN pathway, such as *IFNAR2*, *OAS1*, and *ZC3HAV1/ZAP*, frequently emerged as significant in SARS-CoV-2 studies.

The overlap of gene hits influencing SARS-CoV-2 between our arrayed screen and 15 published whole-genome pooled screens [19,31,64–67,69–71,74,75,77–80] was higher than expected by chance (**S4 Fig**). This finding indicates that genetic screens conducted with different methods (for instance, CRISPR activation or CRISPR KO, pooled or arrayed format) and in various cellular contexts (for instance, Huh-7.5, A549-ACE2, Calu-3, with or without IFN pretreatment) exhibit both specificity and significant overlap. A pathway analysis of the hits from our arrayed screen, alongside hits from pooled screens, is available in **S13 Table**.

We focused on 28 genes identified in both human genetic studies of COVID-19 patients and in functional genetic screens in cell culture, including our own (**Fig 3**). These genes are

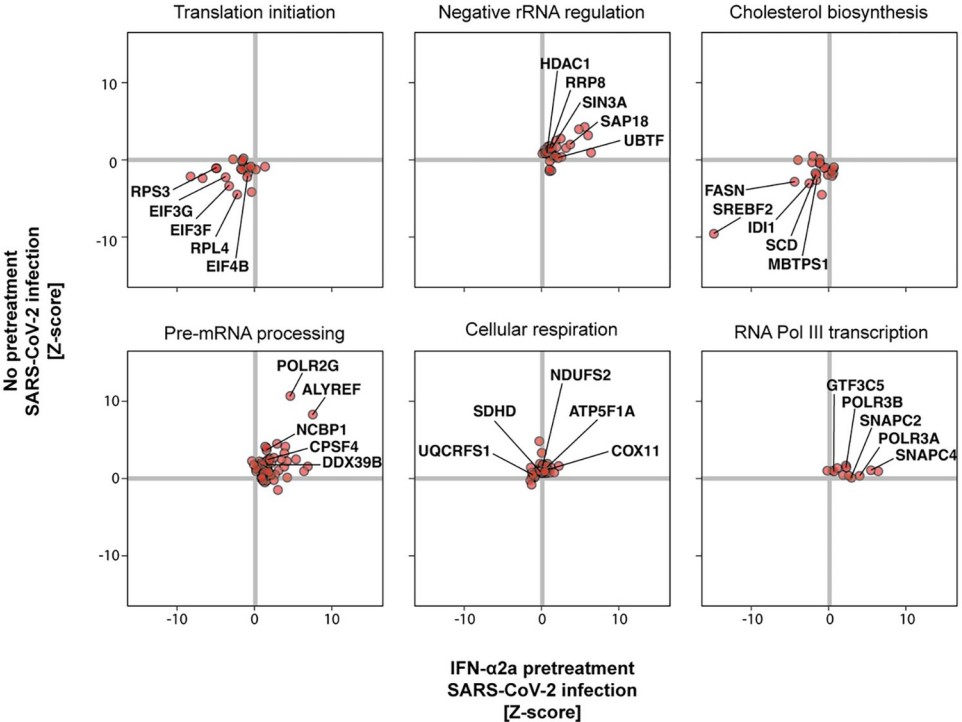

**Fig 2. Cellular pathways influencing SARS-CoV-2 infection.** GSEA was performed on the screen data (**S10 Table**). The individual genes for some of the top pathways are shown here. The full names of the plotted pathways are as follows: translation initiation, "REACTOME EUKARYOTIC TRANSLATION INITIATION"; negative rRNA regulation, "REACTOME NEGATIVE EPIGENETIC REGULATION OF RRNA EXPRESSION"; cholesterol biosynthesis, "REACTOME REGULATION OF CHOLESTEROL BIOSYNTHESIS BY SREBP SREBF"; pre-mRNA processing, "REACTOME PROCESSING OF CAPPED INTRON CONTAINING PRE MRNA"; cellular respiration, "WP ELECTRON TRANSPORT CHAIN OXPHOS SYSTEM IN MITOCHONDRIA"; RNA Pol III transcription, "REACTOME RNA POLYMERASE III TRANSCRIPTION INITIATION FROM TYPE 3 PROMOTER." Pathways are from the Reactome [189] and Wikipathways [191] databases. The data underlying this Figure can be found in **S1 Table**.

likely to have significant physiological relevance and to be well suited for mechanistic studies in cell culture. Among these, the ISG *PLSCR1* stood out, being identified as one of the most potent antiviral genes in our screen (**Fig 1F**). *PLSCR1* variants have been linked to severe COVID-19 in a recent genome-wide association study (GWAS) (listed in **Table 1**) [25,26]. This was attributed to a role of PLSCR1 in regulating the IFN response in COVID-19 patients [26]. Indeed, a pioneering study showed that PLSCR1 potentiates the transcriptional response to IFN-β treatment in human ovarian carcinoma Hey1B cells [24]. However, PLSCR1 surprisingly appeared as a potent SARS-CoV-2 antiviral even in the absence of IFN in our screens, suggesting a cell-intrinsic, IFN-independent function. In other words, baseline levels of PLSCR1 may be sufficient to restrict SARS-CoV-2, and IFN pretreatment could simply enhance this effect by elevating cellular PLSCR1 levels.

## Intrinsic PLSCR1 restricts SARS-CoV-2 independently of the IFN pathway

To better characterize the function of PLSCR1 during SARS-CoV-2 infection, we generated and validated by western blot (WB) PLSCR1 KO bulk Huh-7.5 and A549-ACE2 lines (**S5A Fig**). As observed in the arrayed screen (**S2B Fig**), PLSCR1 KO cells were viable (**S5B Fig**). PLSCR1 depletion increased susceptibility to SARS-CoV-2 independently of IFN pretreatment

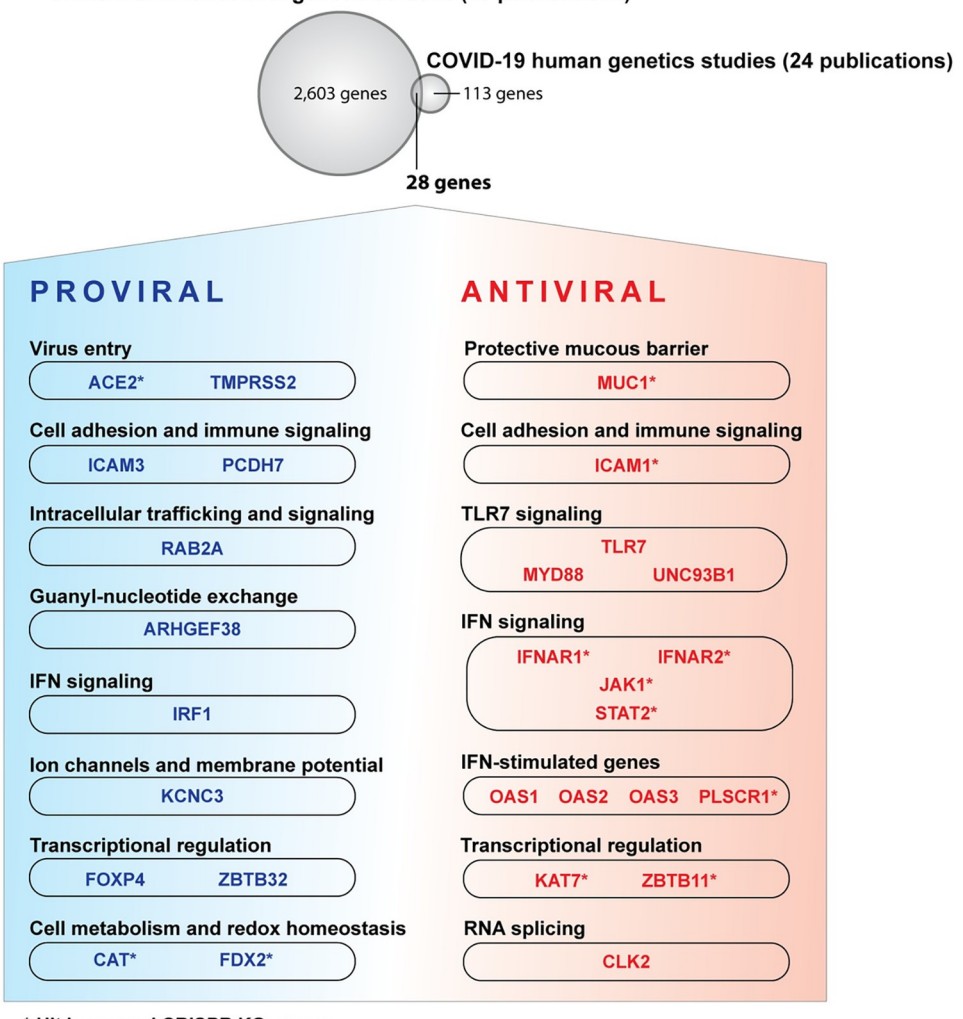

**Fig 3. Human genes significant in human genetics studies on COVID-19 patients and in functional genetic screens in cell culture.** Overlap between genes considered significant by human genetics studies on COVID-19 patients [5,6,8,25,26,81–99] and functional genetic screens for genes influencing SARS-CoV-2 infection in cell culture [11–13,15,18,19,28,31,64–80]. The full list of genes is available in **S11 Table**, and a summary is in **S12 Table**. Please note that not all genes represented here will influence COVID-19 outcomes. We apologize to the many colleagues whose work was not cited and discussed.

(**Fig 4A**). Cell treatment with a JAK-STAT inhibitor, which effectively abrogated IFN signaling, confirmed that intrinsic PLSCR1 limits SARS-CoV-2 infection independently of the IFN signaling pathway (**Fig 4A**). SARS-CoV-2 susceptibility of PLSCR1 KO cells was reversed by the ectopic expression of PLSCR1 (**Figs 4B and S5C**). Interestingly, while PLSCR1 tagged with an N-terminal FLAG tag could rescue, PLSCR1 tagged with a C-terminal FLAG tag could not. The C-terminus of the protein is extracellular, and previous research suggests that this region is important for the protein's scramblase activity and $Ca^{2+}$ binding [117]. It is possible that the addition of this FLAG-tag impaired $Ca^{2+}$ binding, affected PLSCR1's localization at the plasma membrane, or otherwise disrupted the structure of this region, thereby abolishing PLSCR1's antiviral ability. We cocultured PLSCR1 reconstituted cells and PLSCR1 KO cells in the same well and infected them with SARS-CoV-2. A higher proportion of PLSCR1 KO than PLSCR1

**Table 1. PLSCR1 variants associated with severe COVID-19 in a GWAS [25,26].**

| GWAS $p$-value | rsID | Genomic Coordinate (GRCh38) | Nucleotide Change | Gene | Functional Consequence |
|---|---|---|---|---|---|
| $7.52 \times 10^{-7}$ | rs116553931 | chr3:146430956 | C:T | PLSCR2 | intron variant |
| $1.08 \times 10^{-7}$ | rs454645 | chr3:146514682 | C:T | PLSCR1 | Downstream transcript variant |
| $5.43 \times 10^{-8}$ | rs343320 | chr3:146517122 | G:A | PLSCR1 | p.His262Tyr |
| $8.21 \times 10^{-8}$ | rs343318 | chr3:146518204 | T:C | PLSCR1 | intron variant |
| $1.52 \times 10^{-7}$ | rs343317 | chr3:146518374 | A:G | PLSCR1 | intron variant |
| $1.00 \times 10^{-7}$ | rs186910 | chr3:146520241 | A:G | PLSCR1 | intron variant |
| $1.13 \times 10^{-7}$ | rs173150 | chr3:146520256 | A:T | PLSCR1 | intron variant |
| $1.35 \times 10^{-7}$ | rs71302408 | chr3:146520389 | T:C | PLSCR1 | intron variant |
| $7.06 \times 10^{-8}$ | rs343316 | chr3:146521151 | A:G | PLSCR1 | intron variant |
| $4.64 \times 10^{-8}$ | rs343314 | chr3:146522652 | C:T | PLSCR1 | intron variant |
| $7.46 \times 10^{-8}$ | rs343312 | chr3:146522970 | G:A | PLSCR1 | intron variant |

reconstituted cells were positive for SARS-CoV-2 indicating that PLSCR1 acts in a cell autonomous manner (**Fig 4C**). Overall, our data indicate that intrinsic PLSCR1 restricts SARS-CoV-2 in cell culture, even in the absence of IFN. Given that PLSCR1 mRNA is constitutively expressed in SARS-CoV-2 target cells (**S5D Fig**) [118], its intrinsic antiviral function may also be effective in vivo.

## IFN signaling is unaffected by the loss of PLSCR1 in A549-ACE2 and Huh-7.5 cells

PLSCR1 has been shown to potentiate ISG transcription in IFN-treated Hey1B cells [24]. We thus hypothesized PLSCR1 might enhance the type I IFN response in A549-ACE2 and Huh-

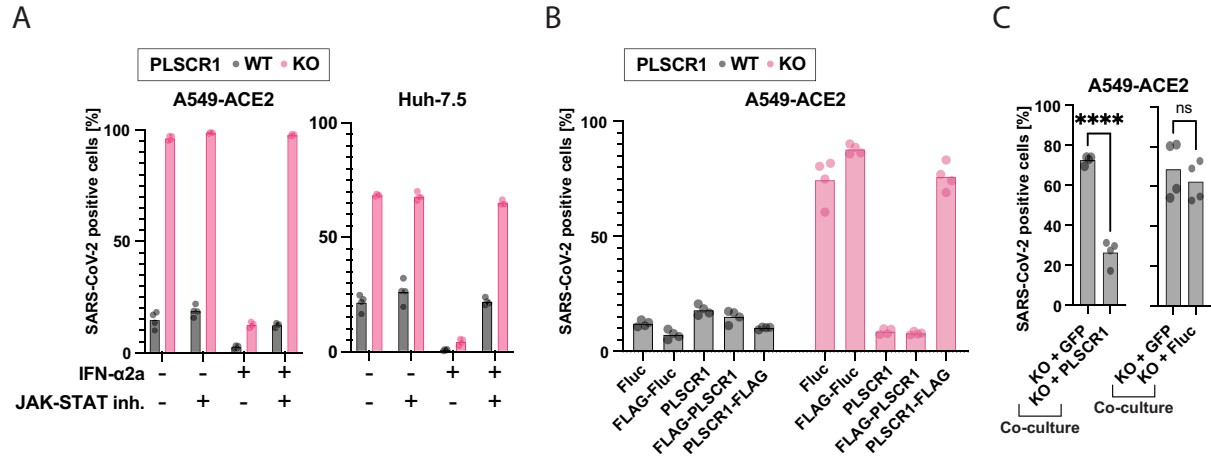

**Fig 4. PLSCR1 is a highly effective anti-SARS-CoV-2 effector ISG contributing to intrinsic immunity in the absence of IFN.** (**A**) Cells were pretreated with a JAK-STAT inhibitor (InSolution 1 μM) for 2 hours, followed by IFN-α2a (10 pM Huh-7.5 or 20 μM A549-ACE2) for 24 hours and were infected with SARS-CoV-2 for 24 hours followed by IF staining for viral N protein. Huh-7.5 infection using an MOI of 0.5 (titer determined by focus forming assay on Huh-7.5 WT cells). A549-ACE2 infection using an MOI of 0.01 (titer determined by focus forming assay on A549-ACE2 WT cells). The percentage of SARS-CoV-2-positive cells is plotted. $n$ = 4 separate wells infected on the same day. (**B**) Cells were reconstituted with the indicated proteins by stable transduction with lentiviruses and then infected as in (**A**). $n$ = 4 separate wells infected on the same day. (**C**) Cells were cocultured as indicated (50:50 mix) and then infected as in (**A**), and the % infection of each cell type was determined. $n$ = 4 separate wells infected on the same day; ****, $p \leq 0.0001$; two-tailed $t$ test. The data underlying this Figure can be found in **S1 Table**. IF, immunofluorescence; IFN, interferon; ISG, IFN-stimulated gene; MOI, multiplicity of infection; PLSCR1, phospholipid scramblase 1; SARS-CoV-2, Severe Acute Respiratory Syndrome Coronavirus 2; WT, wild type.

7.5 cells. We investigated the role of PLSCR1 in the IFN response by infecting Huh-7.5 cells with chikungunya virus (CHIKV), which is unaffected by PLSCR1 KO without IFN (**S6A Fig**). PLSCR1 depletion did not functionally affect the antiviral effects of IFN treatment (**S6B Fig**). Furthermore, IFN treatment induced *OAS1* and *IFI6*, 2 ISGs known to restrict SARS-CoV-2 [11,13,18,95,96,98,99], to a similar extent in both wild type (WT) and PLSCR1 KO cells, indicating that the IFN signaling pathway was unaffected by PLSCR1 depletion (**S6C-S6J Fig**). Finally, PLSCR1 depletion did not alter basal ISG transcription in the absence of IFN (**S7 Fig** and **S14** and **S15** **Tables**).

These findings indicate that PLSCR1 limits SARS-CoV-2 infection independently of the IFN signaling pathway in A549-ACE2 and Huh-7.5 cells.

## PLSCR1 restricts SARS-CoV-2 entry

We hypothesized that PLSCR1 directly targets and inhibits a specific step of the SARS-CoV-2 life cycle. PLSCR1 primarily localized at the plasma membrane in Huh-7.5 cells (**Fig 5A**). Furthermore, PLSCR1 depletion led to increased SARS-CoV-2 foci formation (**Fig 5B and 5C**), and PLSCR1 KO cells did not show increased susceptibility to a SARS-CoV-2 replicon system that bypasses entry (**Fig 5D**) [119]. In contrast, single-cycle, replication-defective human immunodeficiency virus type 1 (HIV-1) particles pseudotyped with SARS-CoV-2 spike showed enhanced entry in PLSCR1-depleted cells (**Fig 5E and 5F**) [120]. These data indicate that PLSCR1 restricts SARS-CoV-2 spike-mediated virion entry. Overexpression of TMPRSS2 lifted the PLSCR1-mediated restriction of authentic SARS-CoV-2 (**Fig 5G-5I**) and of SARS-CoV-2 spike pseudotyped particles (**Fig 5J-5M**), indicating that PLSCR1 primarily restricts the endosomal entry route.

Other ISG products have been described to restrict SARS-CoV-2 entry (**S1 Fig**) [1], notably, CH25H promotes cholesterol sequestration in lipid droplets, decreasing the pool of accessible cholesterol required for virus–cell membrane fusion [28,29]; LY6E blocks virus–cell membrane fusion via currently undetermined mechanisms [121]; nuclear receptor coactivator 7 (NCOA7) overacidifies the lysosome, leading to viral antigen degradation by lysosomal proteases [31,32]; and IFN-induced transmembrane protein 2 (IFITM2) blocks pH- and cathepsin-dependent SARS-CoV-2 virus–cell membrane fusion in the endosome [122]. The aforementioned ISGs were not hits in our CRISPR KO screen for antiviral genes in IFN-treated cells. This may be due to functional redundancies among them. We found that overexpression of any of the aforementioned ISGs restricted SARS-CoV-2 in PLSCR1 KO cells, indicating that they do not need PLSCR1 for their antiviral activity (**S8A Fig**). PLSCR1 is unlikely to require CH25H, LY6E, or IFITM2 for its function as these are expressed at minimal levels in Huh-7.5 cells without IFN pretreatment (**S8B and S8C Fig** and **S4**, **S5**, **S16**, **and S17** **Tables**). We cannot exclude an association between PLSCR1 and NCOA7, although the latter was not hit in our screen.

In addition to SARS-CoV-2, we also evaluated the antiviral activity of PLSCR1 against 10 viruses that utilize endosomal entry: CHIKV, human parainfluenza virus (hPIV), herpes simplex virus 1 (HSV-1), influenza A virus (IAV), human coronavirus OC43 (hCoV-OC43), human coronavirus NL63 (hCoV-NL63), human coronavirus 229E (hCoV-229E), Sindbis virus (SINV), Venezuelan equine encephalitis virus (VEEV), and vesicular stomatitis virus (VSV). Only SARS-CoV-2 showed a notable susceptibility to PLSCR1's inhibitory effects (**S9 Fig**).

One hypothesis for PLSCR1's specificity for SARS-CoV-2 is that it alters the surface levels of its receptor, ACE2. However, flow cytometry on live cells did not show a significant effect of

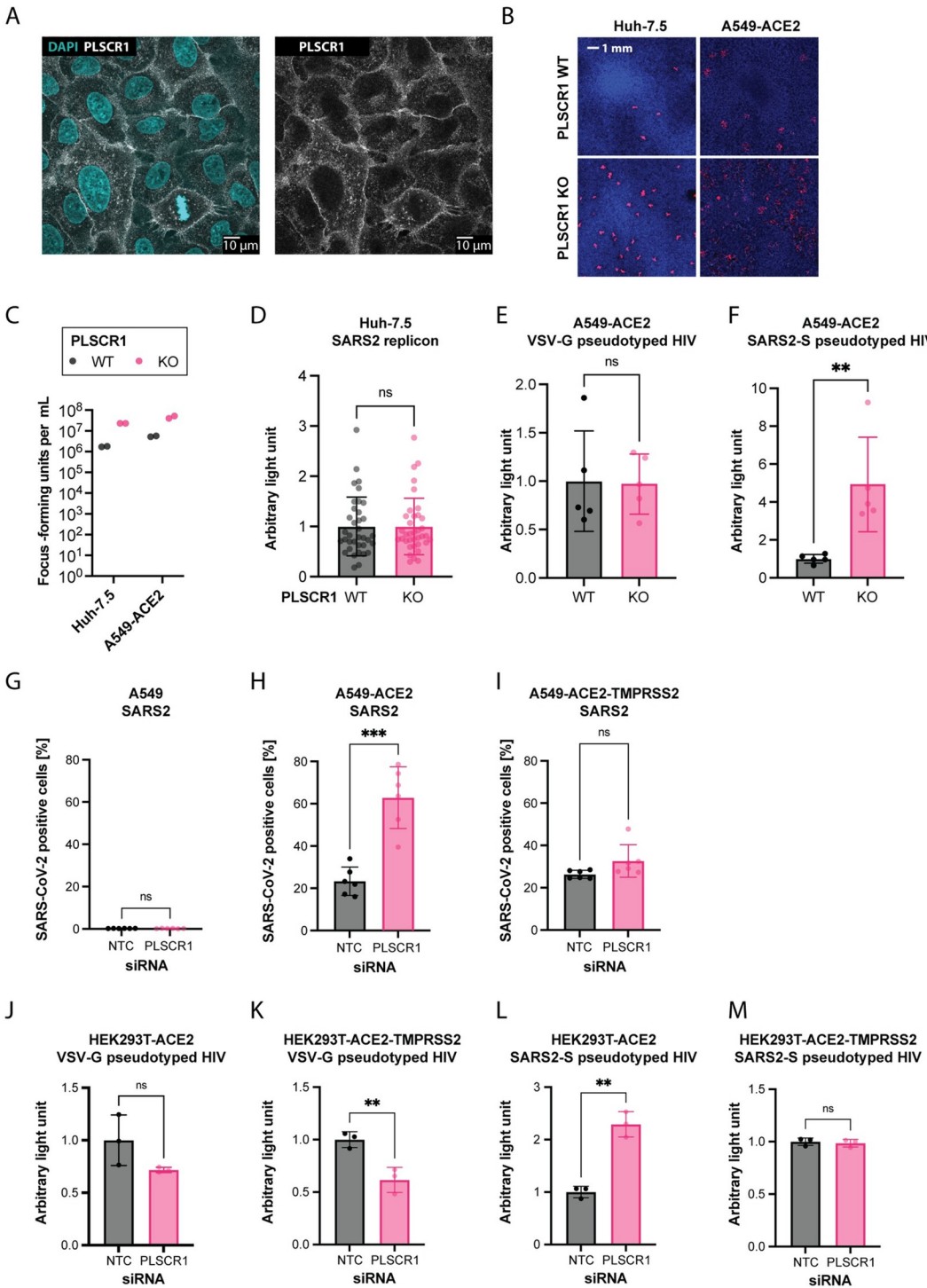

**Fig 5. PLSCR1 restricts spike-mediated SARS-CoV-2 entry. (A)** A549-ACE2 cells were IF stained using an anti-PLSCR1 antibody (white) and Hoechst 33342 nuclear staining (blue) and imaged at 63× magnification on a confocal microscope. (**B**) Focus forming assays: SARS-CoV-2 N IF (red) and Hoechst 33342 nuclear staining (blue) on similarly infected WT or PLSCR1 KO Huh-7.5 and A549-ACE2 cells after 2 and 3 days, respectively. (**C**) Quantification of (**B**) n = 2 separate wells infected on the same day. (**D**) Huh-7.5 WT and PLSCR1 KO cells electroporated with SARS-CoV-2 replicon, which produces a secreted luciferase. Luciferase activity assayed 24 hours after electroporation. *n* = 36 separate wells from a single electroporation event. Error bars represent SD; ns, nonsignificant; two-tailed *t* test. (**E, F**) Transduction of A549-ACE2 cells with an HIV-based replicon expressing the nanoluciferase pseudotyped with VSV-G or SARS-CoV-2 spike, respectively. *n* = 5 separate wells transduced on the same day. Nanoluciferase signal measured 2 dpi. Error bars represent SD; ns,

nonsignificant; **, $p \leq 0.01$; two-tailed $t$ test. (**G-I**) A549 cells WT, expressing ACE2, or expressing ACE2-TMPRSS2 as indicated were transfected with PLSCR1 or NTC siRNAs as indicated for 3 days and infected with SARS-CoV-2 for 1 day. SARS-CoV-2 N was stained by IF, and the percentage of positive cells was determined by imaging. $n = 6$ separate wells infected on the same day. Error bars represent SD; ns, nonsignificant; **, $p \leq 0.01$; ***, $p \leq 0.001$; two-tailed $t$ test. (**J-M**) HEK293T cells expressing ACE2 or ACE2-TMPRSS2, as indicated, were transfected with siRNA targeting PLSCR1, for 3 days and transduced with an HIV-based replicon expressing the nanoluciferase pseudotyped with VSV-G or SARS-CoV-2 spike, as indicated for 2 days. $n = 3$ separate wells transduced on the same day. Error bars represent SD; ns, nonsignificant; **, $p \leq 0.01$; two-tailed $t$ test. The data underlying this Figure can be found in **S1 Table**. ACE2, angiotensin converting enzyme 2; IF, immunofluorescence; KO, knockout; NTC, non-targeting control; PLSCR1, phospholipid scramblase 1; SARS-CoV-2, Severe Acute Respiratory Syndrome Coronavirus 2; TMPRSS2, transmembrane serine protease 2; WT, wild type.

PLSCR1 on ACE2 surface levels in A549-ACE2 cells (**S10 Fig**). The precise mechanism of action of PLSCR1 remains undetermined.

## Recent variants of SARS-CoV-2 are less restricted by PLSCR1

During the COVID-19 pandemic, SARS-CoV-2 variants evolved from the initial strain, showing increased immune evasion and transmissibility [123–125]. To examine if these variants could circumvent the antiviral action of PLSCR1, we infected WT and PLSCR1 KO Huh-7.5 cells with an early strain isolated in July 2020 (NY-RU-NY1, subsequently referred to as "parental") and the Beta (B.1.352), Delta (B.1.617.2), Omicron (BA.5), and Omicron (XBB.1.5) variants. PLSCR1 continued to restrict these later variants when examining the percentage of infected cells 24 hours postinfection (**Fig 6A-6E**).

To determine if the magnitude of PLSCR1 restriction was the same for the parental SARS-CoV-2 strain and recent variants, we plotted the percentage of infection data from **Fig 6A-6E** as a ratio of PLSCR1 WT to KO (**Fig 6F**). Recent SARS-CoV-2 variants showed reduced differences in infection rates between PLSCR1 WT and KO cells than the parental SARS-CoV-2 strain. The diminished difference in sensitivity between PLSCR1 WT and KO cells was most pronounced with Omicron BA.5 and its descendant, XBB.1.5 (**Fig 6F**).

To examine this further, we infected PLSCR1 WT and KO Huh-7.5 cells with approximately 50 focus-forming units (FFU) per well for different SARS-CoV-2 strains (**Fig 6G and 6H**). In line with **Fig 6F,** the data indicate that the difference in virus susceptibility between PLSCR1 WT and KO cells is lower for more recent variants such as Beta (B.1.352) and especially Omicron (XBB.1.5) compared with the parental strain (**Fig 6H**).

Finally, we quantified virus production over an infectious time course for the parental SARS-CoV-2 strain versus Omicron (XBB.1.5) in Huh-7.5 PLSCR1 WT and KO cells (**Fig 6I and 6J**). As expected, parental SARS-CoV-2 replicated with faster kinetics upon PLSCR1 depletion. In contrast, Omicron (XBB.1.5) replication was not affected by PLSCR1 depletion. Our data suggest that PLSCR1 restricts newer SARS-CoV-2 variants less efficiently in Huh-7.5 cells. This could be due to adaptation of the recent variants to directly antagonize PLSCR1 and/or utilize an alternative entry route in Huh-7.5 cells that is both TMPRSS2 independent and invulnerable to PLSCR1.

## Association between PLSCR1 variants and severe COVID-19

PLSCR1 encodes a 318-amino acid protein containing a palmitoylation motif and a transmembrane domain, which regulate its plasma membrane localization, and a nuclear localization signal (NLS) and transcriptional activation domain thought to be important for its nuclear functions (**Fig 7A**) [22,126–131].

A recent GWAS has identified an association between PLSCR1 variants and severe COVID-19 outcomes, reporting an odds ratio of approximately 1.2 and a $p$-value of

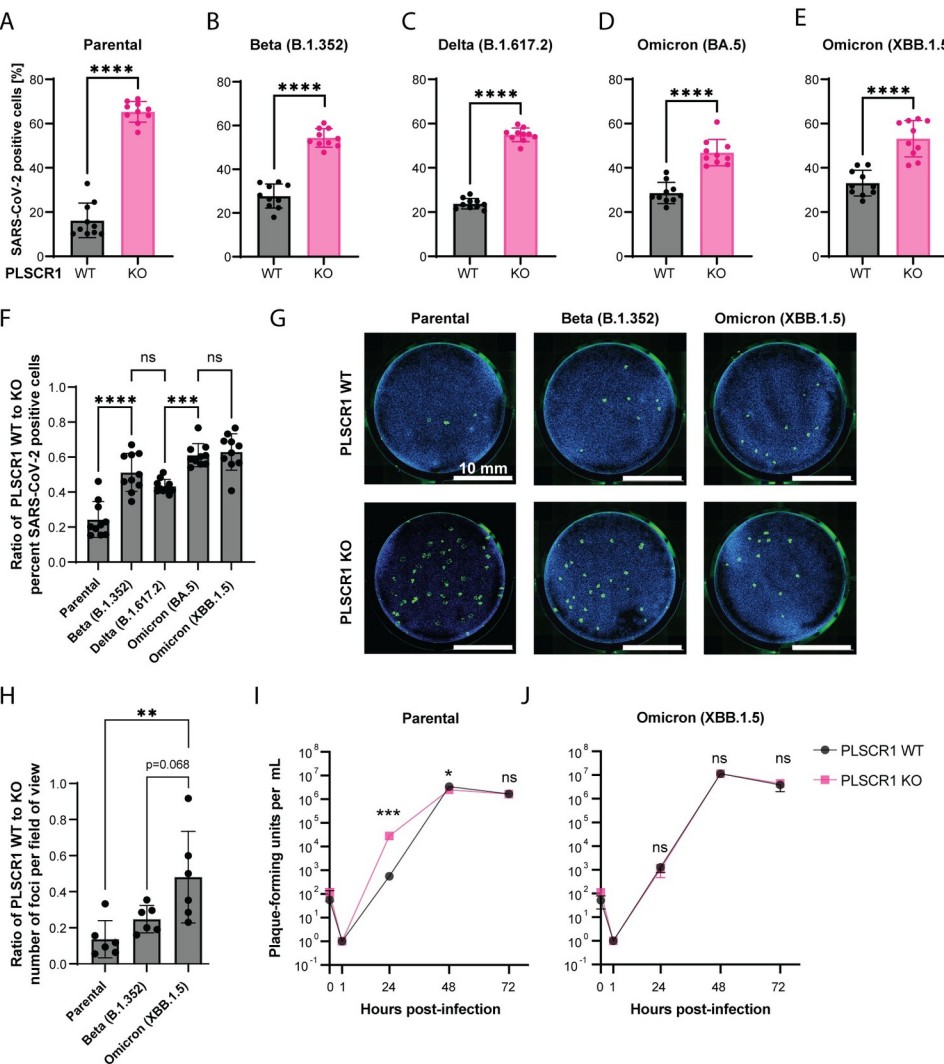

**Fig 6. Newer variants of SARS-CoV-2 are less restricted by PLSCR1.** (**A-E**) Infection of Huh-7.5 cells with SARS-CoV-2 (parental) or its descendant variants, Beta, Delta, Omicron BA.5, and Omicron XBB.1.5 for 24 hours. SARS-CoV-2 N was stained by IF and the percentage of positive cells determined by imaging. $n$ = 10 separate wells infected on the same day. Error bars represent SD; ****, $p < 0.0001$; two-tailed $t$ test. (**F**) Ratio of WT/KO percent infection from (**A-E**). Error bars represent SD. ns, nonsignificant; ***, $p \leq 0.001$; ****, $p \leq 0.0001$; one-way ANOVA. (**G**) Focus forming assay on Huh-7.5 WT and PLSCR1 KO cells infected with approximately 50 FFU of SARS-CoV-2 variants as indicated. FFUs were determined on PLSCR1 KO Huh-7.5 cells. Representative images. (**H**) Foci from (**G**) were counted, and then a ratio of WT-to-KO plotted for each SARS-CoV-2 variant. $n$ = 6 separate wells infected on the same day. Error bars represent SD. **, $p \leq 0.01$; one-way ANOVA. (**I**) Virus production over an infectious time course (growth curve) for the parental SARS-CoV-2 strain. $n$ = 2 to 3 separate wells infected on the same day. Error bars represent SD. ns, nonsignificant, *, $p \leq 0.05$, ***, $p \leq 0.001$, two-tailed $t$ test on the log10-transformed values. (**J**) As in (**I**) with the Omicron (XBB.1.5) strain. The data underlying this Figure can be found in **S1 Table**. FFU, focus-forming unit; IF, immunofluorescence; KO, knockout; PLSCR1, phospholipid scramblase 1; SARS-CoV-2, Severe Acute Respiratory Syndrome Coronavirus 2; WT, wild type.

approximately $10^{-8}$ (**Table 1**) [25,26]. In other words, the GWAS suggests that PLSCR1 has a small but significant effect on severe COVID-19 risks.

GWAS typically identify variants associated with increased odds of a disease, but these variants are not necessarily causative. Among the PLSCR1 variants identified in the COVID-19 GWAS cited above, only p.His262Tyr (rs343320) results in a protein-coding change, located in

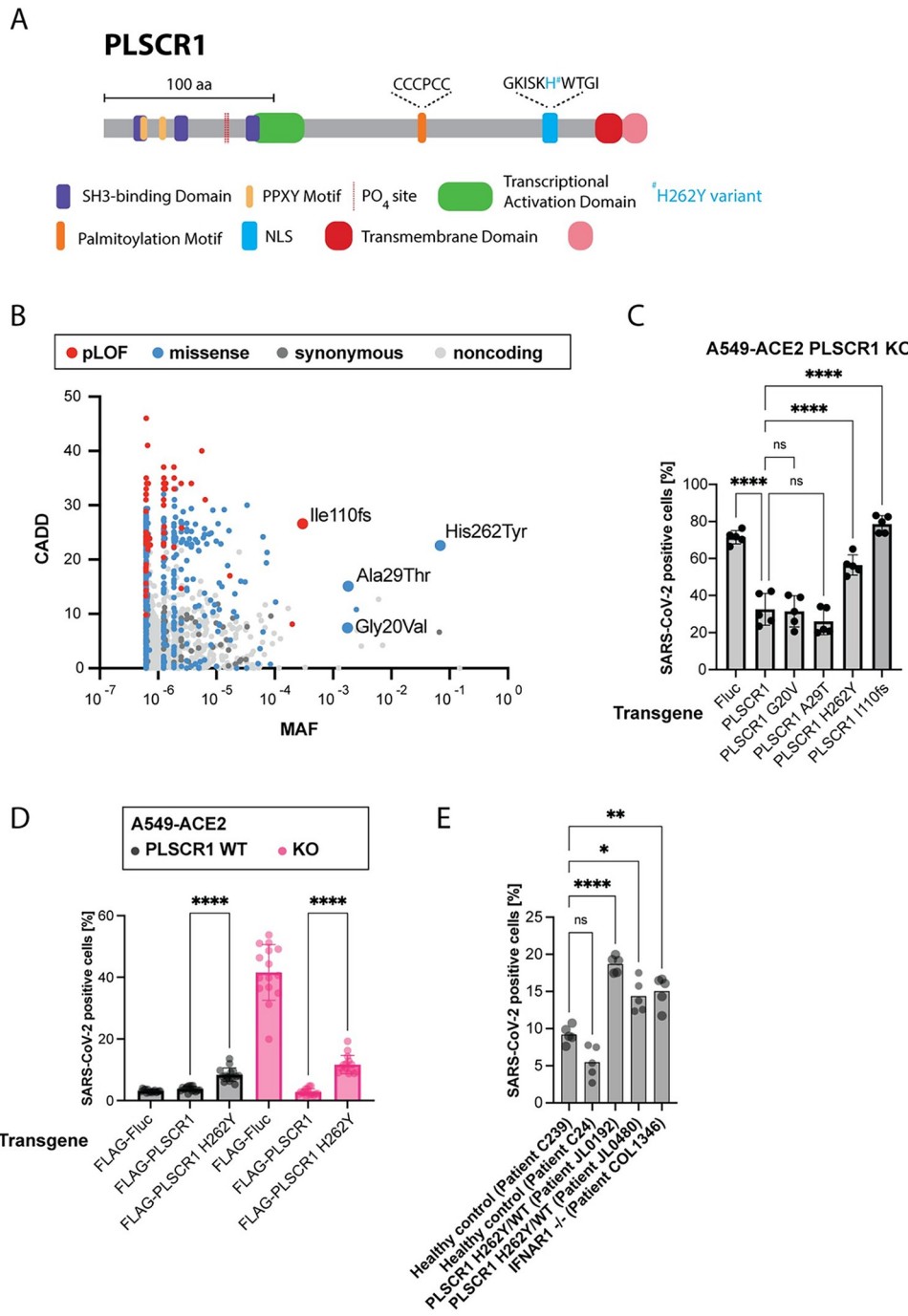

**Fig 7. PLSCR1 p.His262Tyr, which associates with severe COVID-19, leads to higher SARS-CoV-2 infection in cell culture.** (**A**) Protein diagram of PLSCR1. Domain coordinates from UniProt [197]. (**B**) CADD MAF plot showing the common variants of the *PLSCR1* gene reported in gnomAD [132,133]. The plot displays the CADD scores, indicating the predicted deleteriousness of each variant, against the MAF. (**C**) A549-ACE2 PLSCR1 KO cells were transduced to stably and ectopically express the indicated PLSCR1 variants. The cells were then infected for 24 hours with SARS-CoV-2. SARS-CoV-2 N was stained by IF and the percentage of positive cells determined by imaging. $n = 4$ separate wells infected on the same day. Error bars represent SD; **, $p \leq 0.01$; ****, $p < 0.0001$; one-way ANOVA. (**D**) A549-ACE2 cells, WT and PLSCR1 KO, stably expressing N-terminal FLAG-tagged Fluc, N-terminal FLAG-tagged PLSCR1, or N-terminal FLAG-tagged PLSCR1 H262Y mutant and infected with SARS-CoV-2 for 24 hours. SARS-CoV-2 N was stained by IF and the percentage of positive cells determined by imaging. $n = 15$ separate wells infected on the same day. Error bars represent SD; ****, $p \leq 0.0001$; two-tailed $t$ test. (**E**) SV40-Fibroblast-ACE2 cells,

genotype as indicated, infected for 24 hours with SARS-CoV-2. *n* = 8 separate wells infected on the same day. ns, nonsignificant; *, $p \leq 0.05$; **, $p \leq 0.01$; ****, $p \leq 0.0001$; one-way ANOVA. The data underlying this Figure can be found in **S1 Table**. ACE2, angiotensin converting enzyme 2; CADD, Combined Annotation Dependent Depletion; COVID-19, Coronavirus Disease 2019; Fluc, Firefly Luciferase; Fs, frameshift; IF, immunofluorescence; KO, knockout; MAF, minor allele frequency; PLSCR1, phospholipid scramblase 1; SARS-CoV-2, Severe Acute Respiratory Syndrome Coronavirus 2; WT, wild type.

the NLS (**Fig 7A** and **Table 1**). This raises the hypothesis that p.His262Tyr might alter PLSCR1's antiviral function. However, we cannot dismiss the possibility that (i) some noncoding variants identified in the GWAS could influence the regulation of PLSCR1 mRNA, potentially leading to functional outcomes, and (ii) the GWAS might have missed other nonsynonymous variants besides p.His262Tyr that could potentially impact PLSCR1 function.

To assess the relative importance of the p.His262Tyr variant in humans, we examined the full collection of PLSCR1 variants reported in gnomAD, a human population reference database (**Fig 7B**) [132,133]. The p.His262Tyr variant is the most common nonsynonymous variant in this gene, with a minor allele frequency (MAF) of 0.07, MAFmax of 0.1, and 4,348 homozygous individuals in gnomAD. Another variant, resulting in a frameshift mutation, p.Ile110fs (rs749938276), is the most common predicted loss-of-function (pLOF) of PLSCR1 in gnomAD (MAF of 0.0003, MAFmax of 0.01, 1 homozygous individual), indicating that PLSCR1 is not essential.

We aimed to investigate the functional effects of the p.His262Tyr and p.Ile110fs variants in cell culture, along with 2 other relatively common nonsynonymous variants, p.Gly20Val (rs79551579, MAF of 0.002, MAFmax of 0.04) and p.Ala29Thr (rs41267859, MAF of 0.01, MAFmax of 0.04). For this, we stably and ectopically expressed the PLSCR1 variants of interest in PLSCR1 KO A549-ACE2 cells using a lentiviral vector [134]. The variant p.Ile110fs did not rescue PLSCR1 KO, confirming it as a loss-of-function mutation (**Fig 7C**). The p.His262Tyr variant, expressed at similar levels to WT PLSCR1 in this system (**S11A Fig**), behaved as a hypomorphic allele, providing only partial rescue of PLSCR1 antiviral function (**Fig 7C and 7D**). Additionally, introducing PLSCR1 p.His262Tyr into PLSCR1 WT A549-ACE2 cells increased their susceptibility to SARS-CoV-2 infection (**Fig 7D**), suggesting a dominant effect. However, this effect might be attributed to the overexpression of PLSCR1 p.His262Tyr from the transgene, compared to the natural expression levels of PLSCR1 WT from the endogenous locus. To counter this, we examined patient-derived SV40-immortalized fibroblasts expressing ACE2 that were heterozygous for p.His262Tyr. These cells were already present in our collection, originating from a female tuberculosis patient from Turkey (JL0192) and a female herpes simplex encephalitis patient from France (JL0480). The origin of these cells is serendipitous; our goal was not to link these diseases to PLSCR1 but rather to infect cells carrying the His262-Tyr variant with SARS-CoV-2. Cells heterozygous for p.His262Tyr were hypersusceptible to SARS-CoV-2 infection compared to PLSCR1 WT control SV40-fibroblasts (**Fig 7E**), further suggesting that p.His262Tyr is dominant. We cannot formally rule out that the examined SV40-fibroblasts may carry other mutations influencing SARS-CoV-2 infection.

Since the p.His262Tyr mutation is located in the NLS region, we hypothesized that it could impair PLSCR1's nuclear localization. However, PLSCR1 WT was primarily localized in the cytoplasm in untreated A549-ACE2 cells (**Fig 5A**), in IFN-treated cells (**S11B Fig**), and in SARS-CoV-2 infected cells (**S11C Fig**). Similarly, PLSCR1 p.His262Tyr was also enriched in the cytoplasm (**S12 Fig**). This suggests that the palmitoylation motif, known to be dominant over the NLS [129], dictates the cytoplasmic localization of PLSCR1. Therefore, we propose that the p.His262Tyr mutation affects PLSCR1 through a mechanism other than altering its nuclear localization.

Our data collectively highlight PLSCR1's function in restricting SARS-CoV-2 entry in cell culture, thereby clarifying the association between PLSCR1 variants and severe COVID-19 outcomes [25,26]. Future human genetic studies are crucial for determining if certain PLSCR1 variants in the population cause increased risks of severe COVID-19. Our results highlight the variant p.His262Tyr as a potential causative candidate, as it caused increased SARS-CoV-2 infection in cell culture.

## Discussion

Here, we conducted an unbiased arrayed CRISPR KO screen on Huh-7.5 cells infected with SARS-CoV-2. The screen revealed novel aspects of SARS-CoV-2 and IFN biology while also confirming previously known facets. Pathways related to mRNA transcription and maturation were identified as antiviral. This observation may stem from the conflict between the host cell and SARS-CoV-2, where the host attempts to export mRNAs from the nucleus to facilitate anti-viral responses while the virus replicates in the cytoplasm, impeding nuclear export [135–138]. RNA Pol III transcription was specifically essential for the IFN-mediated antiviral response, through mechanisms that are yet to be determined. Interestingly, inborn errors in POLR3A and POLR3C have been previously described in patients with severe varicella zoster virus infections [139]. Cellular respiration was identified as a key IFN-independent antiviral pathway. Further-more, mitophagy was identified as proviral. This may indicate the infected cell's increased demand for energy and ATP to combat the virus. Alternatively, cellular respiration may have other, yet-to-be-identified, IFN-independent antiviral roles. Conversely, translation and choles-terol homeostasis emerged as the foremost proviral pathways. These findings underscore the complex, dualistic nature of the interactions between SARS-CoV-2 and host cells.

Our screen notably identified the ISG zinc-finger antiviral protein (*ZC3HAV1/ZAP*) as a proviral factor in IFN-treated cells. Initially, ZC3HAV1/ZAP gained attention as an antiviral factor that targets the SARS-CoV-2 RNA genome [115] and prevents programmed ribosomal frameshifting [140]. Yet, a recent study demonstrated that ZC3HAV1/ZAP also promotes the formation of SARS-CoV-2 nonstructural proteins 3- and 4-induced double-membrane vesi-cles, essential for virus replication [72]. SARS-CoV-2 may have adapted to exploit certain ISG products, such as ZC3HAV1/ZAP, within the cellular environment it encounters. It is still unclear if the seemingly contradictory roles of ZC3HAV1/ZAP—both proviral and antiviral—are caused by distinct isoforms.

Many other ISG products influenced SARS-CoV-2 infection, PLSCR1 being the most potent restriction factor. PLSCR1 did not influence ISG induction as previously reported [24], but rather inhibited spike-mediated SARS-CoV-2 entry through the endocytic route. Our results corroborate a recent study from Xu and colleagues [19] and provide an explanation for the enrichment for PLSCR1 SNPs observed in a GWAS on severe COVID-19 [25,26].

The molecular mechanisms of PLSCR1-mediated restriction of SARS-CoV-2 entry remain to be elucidated. PLSCR1 could be altering the lipid composition at the contact site between the virus and endosomal membranes, akin to the ISG IFN-induced transmembrane protein 3 (IFITM3) for IAV [141–144]. PLSCR1 was first identified as a $Ca^{2+}$-dependent phospholipid scramblase [23], but it is unclear whether PLSCR1 depletion affects the bidirectional move-ment of phospholipids in vivo [145–147]. A C-terminal FLAG-tag abolished the antiviral abil-ity of reconstituted PLSCR1, possibly by interfering with the function of the $Ca^{2+}$ binding domain. In contrast, inhibiting PLSCR1's phospholipid scramblase activity did not alleviate SARS-CoV-2 restriction [19].

PLSCR1 is one of a few ISGs known to restrict SARS-CoV-2 entry (**S1A Fig**) [1,148], with possibly more yet to be discovered. Future research should investigate the associations and

combinatorial effects between antiviral ISGs, which could lead to key mechanistic insights into their mode of action.

Intriguingly, PLSCR1 specifically restricted SARS-CoV-2 in Huh-7.5 and A549-ACE2 cells, but it did not show similar inhibitory effects on other viruses that enter cells via endocytosis. Future studies will investigate the mechanisms behind this specificity. PLSCR1 has been described to inhibit a range of viruses in various cell lines, such as encephalomyocarditis virus, VSV, Epstein–Barr virus, hepatitis B virus, hepatitis C virus, human cytomegalovirus, HIV-1, human T cell lymphotropic virus type 1, and IAV [24,149–155]. It has been proposed that PLSCR1 directly binds viral proteins and impairs their functions to restrict the non-coronaviruses cited above, reviewed in [156]. However, it seems unlikely that PLSCR1 has evolved to interact directly with such a diverse set of viral proteins. An alternative explanation is that diverse viral proteins convergently evolved to bind PLSCR1 as a mechanism of immune evasion—for example, by altering PLSCR1's subcellular localization. Meanwhile, overexpressing PLSCR1 in cell culture could act like a sponge, absorbing these viral proteins and thereby hindering viral function. We searched for PLSCR1 interactions in 10 SARS-CoV-2 protein interactome studies, relying on ectopic expression of individual viral proteins [100–109], and no interaction was reported in 2 or more independent studies. Two interactions were reported in a single study: (i) PLSCR1-ORF7b [102] and (ii) PLSCR1-ORF8 [105], both by proximity biotinylation, which was less stringent compared to affinity purification-mass spectrometry (AP-MS) and yeast two-hybrid (Y2H) techniques (S13 Fig). To date, there is no strong evidence of a direct interaction between a SARS-CoV-2 protein and PLSCR1, although we cannot rule out that such interactions may occur or even appear in the future as SARS-CoV-2 evolves.

Previous research suggests that Omicron has developed increased resistance to IFN [157,158], a trait associated with its highly mutated spike protein [159]. Here, we show that recent SARS-CoV-2 variants, including Omicron BA.5 and XBB.1.5, exhibit reduced sensitivity to PLSCR1-mediated restriction compared to the New York 2020 strain, which served as a reference in our study (Fig 6). Evasion from PLSCR1 may thus have contributed to Omicron's increased resistance to IFN.

The original SARS-CoV-2 strain can enter cells through both TMPRSS2-dependent fusion near the cell surface and clathrin-mediated endocytosis, with a preference for the former in vivo [38]. ISGs that primarily restrict SARS-CoV-2 endocytosis, such as NCOA7, which perturbs lysosome acidification [160,161], and PLSCR1, may play a role in constraining the original strain to TMPRSS2-dependent entry. Interestingly, the more recent Omicron variant shows both increased evasion from PLSCR1 restriction (Fig 6) and an acquired preference for TMPRSS2-independent entry. This occurs either via endocytosis or TMPRSS2-independent fusion near the cell surface, utilizing metalloproteases to cleave the spike protein [159,162,163]. We postulate that Omicron evolved its entry pathway to evade PLSCR1-mediated restriction while shifting its cell tropism to the upper airways and away from TMPRSS2-expressing cells [162,164–168].

Future research should investigate the following: (i) mechanism(s) by which Omicron evades PLSCR1 restriction and whether this evasion is primarily due to mutations in the spike protein; (ii) the association between PLSCR1-mediated restriction and the distinct entry mechanisms utilized by SARS-CoV-2 variants; and (iii) how various intrinsic factors that restrict virus entry, such as PLSCR1, have influenced SARS-CoV-2 entry routes, evolution, and cell tropism. Understanding these aspects of PLSCR1 restriction could provide mechanistic insight into broad strategies of PLSCR1 evasion employed by both newer SARS-CoV-2 variants as well as viruses from other families that were unaffected by PLSCR1 KO. Subversion of PLSCR1 restriction may serve as a useful immune evasion strategy that could be shared by diverse viruses with longer evolutionary selection than SARS-CoV-2.

Several PLSCR1 variants were enriched in a GWAS on severe COVID-19, with a relatively low odds ratio of approximately 1.2 [25,26]. This modest odds ratio likely reflects the complex redundancies within antiviral defenses, from innate immunity featuring multiple effector ISGs that restrict SARS-CoV-2 [10–15,18,19], to adaptive immunity [169]. For example, we show that the ISGs CH25H [28,29], IFITM2 [122], LY6E [121], and NCOA7 [31,32] still function to restrict SARS-CoV-2 in PLSCR1-depleted cells (**S8 Fig**). However, for these very reasons, the identification of PLSCR1 variants in the GWAS remains noteworthy. Of these enriched variants, only PLSCR1 p.His262Tyr (rs343320) resulted in a protein-coding change. Our findings indicate that p.His262Tyr exhibits a hypomorphic and dominant effect in cell culture, leading to increased SARS-CoV-2 infection. Future research should aim to ascertain whether p. His262Tyr, or potentially other PLSCR1 variants, are directly responsible for elevated risks of severe COVID-19 in patients.

Our findings show that baseline levels of PLSCR1 are effective in limiting SARS-CoV-2 infection. This is in line with other studies where ISGs like *DAXX* and *LY6E* were shown to inhibit SARS-CoV-2 independently of IFN [18,121,170]. mRNA-seq analyses of Huh-7.5 cells and primary human hepatocytes, as well as data from the GTEx consortium on various human tissues [49,171], revealed that many ISGs are constitutively expressed, even without IFN stimulation (**S14 Fig**). This supports the idea that the IFN-induced antiviral state results more from enhanced expression of antiviral genes rather than a binary ON/OFF switch. In future studies, it will be interesting to explore whether intrinsically expressed ISGs also carry out cellular functions beyond pathogen defense.

## Materials and methods

### Plasmids, oligos, and primers

The plasmids, gene fragments, and primers used in this study are listed in **S18, S19, and S20 Tables**, respectively.

### Cell lines

Huh-7.5 (human hepatocellular carcinoma; *H. sapiens*; sex: male) [172], Huh-7.5-Cas9 [31], A549-ACE2 (human lung carcinoma; *H. sapiens*; sex: male; generously provided by the laboratory of Brad R. Rosenberg) [11], A549-ACE2-TMPRSS2 [31], Lenti-X 293T (*H. sapiens*; sex: female: Takara, cat. #632180), Caco2 (*H. sapiens*; sex: male; ATCC, cat. HTB-37), Vero E6 (*Chlorocebus sabaeus*; sex: female; kidney epithelial cells, ATCC cat. #CRL-1586), BHK-21 (*Mesocricetus auratus*; sex: unspecified; kidney fibroblasts) cells, and SV40-Fibroblasts were cultured in Dulbecco's Modified Eagle Medium (DMEM; Fisher Scientific, cat. #11995065) supplemented with 0.1 mM nonessential amino acids (NEAA; Fisher Scientific, cat. #11140076) and 10% fetal bovine serum (FBS; HyClone Laboratories, Lot. #KTH31760) at 37°C and 5% $CO_2$. All cell lines tested negative for mycoplasma.

### Virus stocks

**CHIKV-181/25-mKate2:** The infectious clone was a kind gift from Mark Heise (University of North Carolina, United States of America). The clone was derived from the live attenuated strain 181/25 [173,174]. Mark Heise's lab engineered a fluorescent reporter of this infectious clone by inserting the mKate2 fluorescent protein in-frame within the structural polyprotein. The fluorescent reporter is immediately downstream of the capsid open-reading frame and duplicated E3 cis-acting sequence. Downstream of the mKate2 reporter are the full-length E3, E2, 6K, and E1 open-reading frames. The infectious clone DNA (quantity 20 μg) was linearized

with NotI HF at 37°C overnight. Complete digestion was confirmed by running a sample of the digested DNA on a 1% agarose gel. After confirmation, linearized DNA was cleaned via phenol-chloroform extraction, and then ethanol precipitated. The precipitated DNA was resuspended in 20 μL of RNase-free $H_2O$ and in vitro transcribed with an SP6 mMessage mMachine In Vitro Transcription Kit (Thermo Fisher, cat. AM1340). The generated RNA was electroporated into $1.2 \times 10^7$ BHK-21 cells, and the produced virus was harvested once approximately 80% of the electroporated cells lifted or showed signs of cytopathic effects and 100% of the cells were positive for mKate2 signal. The titer of the virus was $8.5 \times 10^6$ FFUs on Huh-7.5 cells.

**hCoV-229E:** was generously provided by Volker Thiel (University of Bern) and amplified at 33°C in Huh-7.5 cells as in [31].

**hCoV-NL63**: was obtained from BEI Resources (NR-470) and amplified at 33°C in Huh-7.5 cells as in [31].

**hCoV-OC43**: was obtained from ZeptoMetrix (cat. #0810024CF) and amplified at 33°C in Huh-7.5 cells as in [31].

**hPIV3-GFP** [175]: stock (based on strain JS) grown in VeroE6 cells as in [176].

**HSV-1-GFP**: stock made by passage on VeroE6 cells. About $2 \times 10^7$ cells seeded in a T175 flask were infected at an MOI of 0.01 PFU/ml of HSV-1-GFP virus engineered and provided by Ian Mohr [177]. After a 1-hour incubation at 37°C, the inoculum was removed, and 20 ml of DMEM supplemented to contain 10% FBS and NEAA was added. Cells were incubated at 37°C for 24 hours or until CPE was evident. Cell supernatant containing progeny virus was harvested and titrated on Vero E6 cells (2.4% avicel, fix 2 dpi) at $2.4 \times 10^8$ PFU/ml.

**IAV WSN (H1N1):** was generated in MDCK cells. Cells were inoculated at MOI 0.01 in DMEM supplemented with NEAA, 0.2% BSA, 0.1% FCS, 50 mM HEPES, and 1 μg/ml TPCK-trypsin. Virus-containing culture supernatant was harvested at 52 hours postinfection and cleared by centrifugation.

**SARS-CoV-2:** Unless otherwise stated, the isolate SARS-CoV-2/human/USA/NY-RU-NY1/2020 was used in this study [178]. The virus was sourced from the saliva of a deidentified patient in New York City, collected on July 28, 2020. Its sequence is publicly accessible (GenBank OM345241). The virus isolate was amplified in Caco-2 cells. The passage 3 stock employed had a titer of $3.4 \times 10^6$ PFU/ml, as measured on Vero E6 cells using a 1% methylcellulose overlay, according to previously described methods [179]. The Beta (B.1.351), Delta (B.1.617.2), Omicron BA.5, and Omicron XBB.1.5 variants were obtained from BEI resources (cat. # NR-54008, NR-55611, NR-58616, and NR-59104, respectively), amplified in Vero E6 cells engineered to stably express TMPRSS2, and titer was determined as described above.

**SINV Toto1101** [180]: Expressing an nsP3-mScarletI fusion reporter was generated by cloning the sequence encoding mScarletI in frame into a unique SpeI restriction site in the pToto1101 infectious clone plasmid as previously described [181]. In vitro transcribed, capped RNA was generated from the pToto1101-nsP3-mScarletI plasmid (Invitrogen mMessage mMachine SP6 kit, AM1340) and electroporated into BHK-J cells, a derivative of BHK-21 cells (ATCC, CCL-10) as previously described [181]. Centrifuge-clarified supernatants were aliquoted and stored at −80°C, 24 hours post electroporation. BHK-J cells were cultured and virus stocks generated in MEM supplemented with 7.5% FBS.

**VEEV-dsEGFP** [16,182]: The infectious clone plasmid was linearized (MluI) and transcribed in vitro using an mMessage mMachine SP6 transcription kit (Ambion). BHK-21 cells were electroporated with viral RNA, and supernatant containing progeny virus was harvested after incubation at 37°C for 30 hours or until CPE was evident. Virus was titrated by plaque assay on BHK-21 cells (2.4% avicel, fix 2 dpi). BHK-21: $1.45 \times 10^9$ FPU/ml.

**VSV-GFP** [183]: grown in BHK-21 cells as in [176].

**YFV 17D:** was generated via transfection of Huh-7.5 with in vitro transcribed RNA from pACNR-FLYF-17D plasmid as described in [176].

## mRNA-seq

**mRNA-seq on SARS-CoV-2-infected cells.    Cell culture and infection:** Approximately 75,500 Huh-7.5 cells or 150,000 Calu-3 cells were seeded in each well of a 12-well plate with 1 mL media. Media: DMEM with 5% FBS and 1% NEAA for Huh-7.5 cells or EMEM (ATCC, 30-2003) with 10% FBS for Calu-3 cells. The next day, cells were infected by removing 500 μL of media and adding 500 μL of media with SARS-CoV-2 strain USA-WA1/2020 (BEI Resources, NR-52281) at 5,000 PFU/well (virus titer determined in Huh-7.5 cells). After 1 day, the wells were washed with PBS and cells were harvested in 1 mL TRIzol (Invitrogen, cat. 15596-018). $N$ = 3 replicates (separate wells) per sample.

**RNA extraction:** MaXtract High Density tubes (2 mL, Qiagen, 129056) were centrifugated at 12,000 to 16,000 × $g$ for 20- to 30-second centrifugation. A volume of 750 μL TRIzol-prepared sample was combined with 150 μL chloroform in these tubes and hand-shaken vigorously. Phase separation was accomplished by centrifugation at 1,500 × $g$ for 5 minutes at 4˚C. The aqueous phase was then mixed with 400 μL ethanol 95% to 100% in a separate tube. These preparations were then transferred to Zymo Research RNA clean and concentrator-25 kit columns (Zymo Research, cat. R1018) and subjected to multiple wash and centrifugation steps as recommended by the manufacturer. An in-column DNase I treatment was performed using Qiagen DNase (Qiagen, 79254). Finally, RNA was eluted with 50 μL DNase/RNase-Free water and stored at −80˚C.

**Sequencing:** Poly A-enriched libraries were made using the TruSeq stranded mRNA LT kit (Illumina, Cat# 20020594) and sequenced on a NovaSeq SP with PE150 read length.

**mRNA-seq on IFN-treated cells.    Cell culture and treatment: A549 cells**: Approximately 300,000 cells were seeded in each well of a 6-well plate with 3 mL of DMEM supplemented with 10% FBS and 1% NEAA. The following day, the media was aspirated and replaced with 2 mL of DMEM with or without 0.5 nM IFN-α2a (PBL, cat. 11101-2), and the cells were incubated at 37˚C. After 24 hours, the cells were harvested in 500 μL TRIzol. $N$ = 3 replicates (separate wells treated on the same day) per sample. **Calu-3 cells**: Around 200,000 cells were seeded in each well of a 12-well plate with 1 mL media. Two days later, the media was replaced with EMEM (ATCC, cat. 30-2003) with 10% FBS with 0.5 IFN-α2a (PBL, cat. 11101-2) and incubated at 37˚C. Cells were harvested in 500 μL TRIzol, 24 hours later. $N$ = 3 replicates (separate wells treated on the same day) per sample. **Huh-7.5 cells**: A total of 75,500 cells were seeded in each well of a 12-well plate with 1 mL media. Two days later, the media was replaced with 1 mL of DMEM with 5% FBS, 1% NEAA with 0.5 nM IFN-α2a (PBL, cat. 11101-2) and incubated at 37˚C. Cells were harvested in 500 μL TRIzol, 24 hours later. $N$ = 3 replicates (separate wells treated on the same day) per sample.

**RNA extraction** as described above.

**Sequencing:** Poly A-enriched libraries were made using the NEBNext Ultra II RNA Library Prep Kit for Illumina (NEB, cat. E7770) and sequenced on a NovaSeq SP with PE150 read length.

**mRNA-seq on PLSCR1 KO cells.    Cell culture and CRISPR KO:** Approximately 30,000 Huh-7.5 cells were seeded in 5 wells of a 24-well plate with 480 uL media. The cells were reverse transfected with 120 μL of a transfection mixture composed of 250 nM of pooled anti-PLSCR1 or nontargeting Edit-R crRNAs from Horizon Discovery (cat. CM-003729-01-0002, CM-003729-02-0002, CM-003729-03-0002, and CM-003729-04-0002 or U-007501-01-05, U-007502-01-05, U-007503-01-05, and U-007504-01-05, respectively), which had been resuspended with an equimolar amount of Edit-R tracrRNA (Horizon, cat. U-002005-20) and a

1:200 dilution of Dharmafect 4 (Horizon, cat. T-2004-01). The following day, the media was changed, and the cells were progressively scaled up to a 6-well plate over the next 4 days. When the cells were confluent in the 6-well plate, the media was removed from 4 of the wells. They were then washed with 1× PBS (cat. 14190-144) and lysed with 1 mL TRIzol (Life Technologies, cat. 15596-018) for 5 minutes at room temperature before transferring to an Eppendorf tube and freezing at −80°C to await RNA extraction. The remaining well was lysed with 300 μL of RIPA buffer (Thermo cat. 89900) supplemented with 1× protease inhibitor (Thermo cat. 87786) and 1× EDTA and prepared for WB as described below, in the "Western blots" section.

**RNA extraction** as described above.

**Sequencing:** Poly A-enriched libraries were made using the NEBNext Ultra II RNA Library Prep Kit for Illumina (NEB, cat. E7770) and sequenced on a NovaSeq SP with PE150 read length.

**mRNA-seq analysis.** mRNA-seq reads were first quality-filtered and adapter-trimmed using Trim Galore with parameters -q 20 -e 0.1 –length 20 –paired and Cutadapt. Reads were then mapped to the human genome GRCh38 or to a combined SARS-CoV-2 MN985325.1/ human genome GRCh38 using STAR [184] with settings including –runThreadN 8 –outFilter-MultimapNmax 1 –twopassMode Basic. Feature counting was performed using the feature-Counts function from the Rsubread package [185], with strandness specified depending on the sequencing and other parameters as default. The resulting counts were imported into a DESeq-DataSet object using the DESeq2 package [186] with a design formula of ~Group. Size factors were estimated and normalized counts were extracted and saved. Differential expression analysis was performed using DESeq with the created DESeqDataSet object, contrasted by sample groups, cooksCutoff and independentFiltering disabled, and otherwise default parameters.

## TMPRSS2 western blot

Cell lysates were prepared in RIPA buffer (150 mM NaCl, 1% NP-40, 0.5% DOC, 0.1% SDS, 50 mM Tris-HCl [pH 7.4] with addition of Halt Protease and Phosphatase Inhibitor Cocktail [Thermo Fisher: 78440]) and incubated on ice for 30 minutes. Lysates were clarified by centrifugation at 16,000 × $g$ for 10 minutes at 4°C. Protein concentration was determined by bicinchoninic (BCA) protein assay (Pierce BCA Protein Assay Kit, Thermo Fisher Scientific: 23227), and samples were resolved on NuPAGE 4% to 12% Bis-Tris gels (Invitrogen) followed by transfer onto 0.4 μm nitrocellulose membranes. Membranes were incubated in blocking buffer (TBS + 5% milk) and incubated with primary antibody prepared in blocking buffer with 0.1% Tween-20. Membranes were washed 3× with TBS + 0.1% Tween-20 and incubated with LI-COR (Lincoln, Nebraska, USA) IRDye 680RD or IRDye 800CW secondary antibodies. Membranes were imaged using an Azure (Dublin, California, USA) 600 imaging system.

## Unbiased arrayed CRISPR KO screening

**Screen overview.** The content of each gRNA 384-well plate constituting the whole-genome library (61 library plates total) was transfected to 16 assay 384-well plates (976 assay plates total). Positive and negative control gene gRNAs were incorporated into vacant wells of each assay plate as described below. Huh-7.5-Cas9 cells were subsequently seeded into these assay plates. The 16 assay plates served as replicates for 3 distinct experimental conditions: 4 replicates for mock treatment followed by mock infection, 5 replicates for IFN-α2a treatment followed by SARS-CoV-2 infection, and 7 replicates for mock treatment followed by SARS-CoV-2 infection. Each day, 3 library 384-well plates were processed, along with their corresponding 48 assay plates. The full gRNA library, distributed across 61 384-well plates, was completed over a span of 21 days. For each set of plates, cell seeding was conducted on day 0, IFN-α2a treatment on day 4, SARS-CoV-2 infection on day 5, and cell fixation on day 6.

**gRNA library preparation.**    A 0.1-nmol Edit-R Human Whole Genome crRNA Library (Horizon, cat. GP-005005-01) containing 4 crRNAs per gene and 1 gene per well (total 0.1 nmol crRNA/well) was resuspended in 80 μL of a 1.25-μM tracrRNA (Horizon, cat. U-002005-1000) 10 mM Tris-HCL (pH 7.4) solution to create a 1.25-μM gRNA solution. The library was then aliquoted in 10 mM Tris-HCL (pH 7.4) in several 96-well plate and 384-well plate copies using a Tecan Freedom EVO liquid handler. A single-use library copy containing a 40-μL/well of a 312.5-nM gRNA solution in the 384-well plate format was used in this study.

**gRNA reverse transfection (day 0).**    In each well of the 384-well assay plates, 40 μL of a transfection solution was prepared by combining 2% DharmaFect-4 transfection reagent (Horizon, cat. GP-T-2004-07A) in Opti-MEM (Gibco, cat. 31985070). This was added to 40 μL of a 312.5-nM gRNA library using a Thermo Fisher Multidrop Reagent Dispenser, yielding an 80-μL/well transfection mixture. The mixture was left to incubate at room temperature for 20 minutes. Simultaneously, assay plates were preloaded with 11 μL/well of serum-free media, which was formulated from DMEM, 1× Antibiotic-Antimycotic solution (Gibco, cat. 15240-062), and 1× NEAA, dispensed via a Thermo Fisher Multidrop Reagent Dispenser. Subsequently, 4 μL/well of the transfection mixture was dispensed into each of the assay plates (16 assay plates per library plate) using a Tecan Freedom EVO liquid handler. During this time, Huh-7.5 cells were prepared in media containing 25% FBS, 1× Antibiotic-Antimycotic solution, and 1× NEAA. A volume of 10 μL cells/well was added to the assay plates, again using a Thermo Fisher Multidrop Reagent Dispenser. Ultimately, each well contained 1,250 cells in a 25-μL final volume, with a composition of 25 nM gRNA, 10% FBS, 0.8× Antibiotic-Antimycotic, and 0.8× NEAA. Plates were then span at 200 g for 5 minutes. To minimize evaporation, plates were sealed with Breathe-Easy sealing membranes (Sigma-Aldrich, cat. Z380059) and placed in humid chambers constructed from a 245 mm × 245 mm dish containing a paper towel moistened with 15 mL of 1× Antibiotic-Antimycotic solution. Four assay plates were placed in each humid chamber and incubated at 37°C.

**IFN-α2a treatment (day 1).**    Each well received 5 μL of IFN-α2a (PBL, cat. 11101-2) in media (DMEM, 20% FBS, 1× Antibiotic-Antimycotic solution, 1× NEAA), using a Thermo Fisher Multidrop Reagent Dispenser, for a final concentration of 1 pM IFN-α2a in a final volume of 30 μL. Plates were then span at 200 g for 5 minutes and incubated at 37°C.

**SARS-CoV-2 infection (day 5).**    Each well received 212.5 PFU SARS-CoV-2 virus (titer determined on Vero E6 cells; see Virus stocks section above) diluted in 5 μL of media (DMEM, 20% FBS, 1× Antibiotic-Antimycotic solution, 1× NEAA) for a final volume of 35 μL in the BSL3. Plates were then span at 200 g for 5 minutes and incubated at 37°C.

**Fixing (day 6).**    Each well received 50 μL of 20% neutral buffered formalin (Azer Scientific, cat. 20NBF-4-G) and plates were incubated overnight. The formalin mixture was then removed and each well received 50 μL of PBS.

**IF staining.**    For IF staining of SARS-CoV-2-infected cells in the arrayed CRISPR KO screen (**Fig 1D-1F**), as well as some focused experiments (**Figs 4**, **5G-5I**, **7C**, **7E**, **S8**, and **S9F**), the following solutions were prepared for both 96-well plate (96-wp) and 384-well plate (384-wp): PBS (phosphate buffered saline), Perm Solution: comprised of PBS with an added concentration of 0.1% Triton X100, Blocking Solution: PBS was mixed with 1% BSA. This solution was prepared a day in advance and filtered before use, PBST: PBS with 0.1% of Tween 20, Primary Antibody Solution: Genetex anti-SARS-CoV-2 N poly rabbit antibody (GTX135357) at a dilution of 1:3,000, Secondary Antibody Solution: AF647 anti-rabbit antibodies at a dilution of 1:3,000 and Hoechst 33342 (10 mg/ml) at 1:10,000. Plates were stained on a Biotek EL406 Microplate Washer Dispenser using the following steps: (1) Priming: The washer was primed with 200 ml of each buffer: PBS, Perm Solution, Blocking Solution, and PBST. (2) First Washing Phase: Contents of the plates were aspirated. Plates were then washed

with 50 μL/well (384-wp) or 200 μL/well (96-wp) of Perm Solution, followed by a slow shake for 3 seconds. (3) Permeabilization: A delay of approximately 1 minute was implemented for permeabilization, in addition to the time required to process all the plates (around 1 minute per plate). (4) Second Washing Phase: Plates were washed with 50 μL/well (384-wp) or 200 μL/well (96-wp) of PBS. Subsequently, 50 μL/well (384-wp) or 200 μL/well (96-wp) of Blocking Solution was added to the plates, followed by a slow shake for 3 seconds. (5) Blocking, auto-clean, and Primary Antibody Priming: The washer was set to undergo an autoclean cycle with PBS for 30 minutes. Simultaneously, the syringe containing the Primary Antibody Solution was primed with 16 ml. (6) Third Washing Phase and First Antibody Dispensing: After aspi-rating the contents of the plates, 15 μL/well (384-wp) or 60 μL/well (96-wp) from the Primary Antibody Solution was added, followed by a slow shake for 3 seconds. (7) Primary Antibody Incubation, autoclean, and Secondary Antibody Priming: The washer was subjected to another autoclean cycle using PBS for 2 hours and 5 minutes. The syringe containing the Secondary Antibody Solution was primed with 16 ml during this period. (8) Fourth Washing Phase and Second Antibody Dispensing: Plates were washed with 50 μL/well (384-wp) or 200 μL/well (96-wp) of PBST, followed by a 2-second slow shake and aspiration. Then, 15 μL/well (384-wp) or 60 μL/well (96-wp) from the Secondary Antibody Solution was added, accompa-nied by a 3-second slow shake. (9) Secondary Antibody Incubation and autoclean: An auto-clean cycle with PBS was initiated and lasted for 1 hour. (10) Final Washing Phase: Plates were washed with 50 μL/well (384-wp) or 200 μL/well (96-wp) of PBST. This was followed by 2 con-secutive washes with 50 μL/well (384-wp) or 200 μL/well (96-wp) of PBS, incorporating a 2-second slow shake in each cycle. Finally, plates were left with 50 μL/well (384-wp) or 200 μL/well (96-wp) of PBS.

**Imaging.**    Plates were imaged with an ImageXpress micro-XL and analyzed with MetaX-press (Molecular Devices).

**Analysis.**    Analysis was conducted in R.

**Data omission**: We excluded 5 library plates, constituting 8% of the total library, due to insufficient infection levels for accurate quantification.

**Normalization**: Two variables were subject to normalization—percentage of SARS-CoV-2 positive cells and the count of nuclei. The normalization steps were applied separately for the 3 screening conditions: mock treatment followed by mock infection, IFN-α2a treatment fol-lowed by SARS-CoV-2 infection, and mock treatment followed by SARS-CoV-2 infection. Data were first Z-scale normalized within assay plates:

$$Scale(x) = \frac{x - mean(x)}{sd(x)}$$

and then Z-scale normalized per row and per column to remove any spatial effects.

**Statistics:** A robust statistic accounting for technical and biological variability was applied using the below formula within the replicates of each gene:

$$Stat\ score(x) = \frac{mean(x)}{sd(x)}$$

This statistic was further standardized by Z-scaling across all genes to produce our final z-score.

**Exclusion of genes influencing cell proliferation:** A total of 224 genes with nuclei count z-score $\geq 2$ and 388 genes with nuclei count z-score $\leq 2$ in the mock treatment followed by mock infection condition were deemed to influence cell proliferation and excluded from sub-sequent analyses.

**Exclusion of genes not expressed in cell lines of interest (A549, Calu-3, Huh-7.5 cells) and in human lung cells.** Expression data from cell lines from [47,48]. Expression data from tissues from [49]. Genes were considered expressed if they had at least 1 read count within exons.

## Gene set enrichment analysis

For the pathway analysis in **S2E Fig**, we leveraged the FGSEA package [187] to perform GSEA using gene sets found in the Molecular Signatures Database (MSigDB) [188]: Reactome [189], KEGG [190], Wikipathways [191], Pathway Interaction Database [192], and Biocarta [193]. The analysis was conducted separately for 2 conditions: IFN-α pretreated SARS-CoV-2 infection and non-pretreated SARS-CoV-2 infection. We attributed a score to each pathway for both conditions:

$$Score = -log10(padj) \times sign(Normalized\ Enrichment\ Score)$$

Each pathway was then attributed to 1 of 9 quadrants (as in **Fig 1F**) based on its score in the IFN-α-pretreated condition (axis x) versus non-pretreated condition (axis y), using $padj \leq 0.05$ as a cutoff.

## Compilation of published large-scale omic studies on SARS-CoV-2

As a rule, we listed the genes classified as "hits" by the authors of the respective studies. Below are some exceptions or clarifications:

**Functional genetic screens.  Baggen et al. [80]:** We used the "low stringency adjusted" analysis in Suppl Table 11. Proviral: p_value_neg $\leq$ 0.05 and log2 FC $\geq$ 1. Antiviral: p_value_pos $\leq$ 0.05 and log2 FC $\leq$ -1. We also used "High stringency" analysis in Suppl Table 7. Proviral: Gene is TMEM106B (log2 FC = 3.8 and p_value_neg = 0.08) or p_value_neg $\leq$ 0.05 and log2 FC $\geq$ 1 (no gene matched this criteria). Antiviral: p_value_pos $\leq$ 0.05 and log2 FC $\leq$ -1 (no gene matched this criteria). **Biering et al. [79]:** in Supplementary Table 1, in Tab 1: LOF-enriched screen analysis, for proviral genes, we used FDR $\leq$ 0.05. In Tab 2: GOF-depleted screen analysis, proviral: FDR $\leq$ 0.05. In Tab 3: GOF-enriched screen analysis, for antiviral genes, we used FDR $\leq$ 0.05. **Chan et al. [78]:** in Multimedia component 6, for Vero E6 (T16); UM-UC-4 (T23); HEK293+A+T (T12); HuH-7 (T15) and Calu-3 (T43), we considered gene as hits at FDR < 0.1 (as in Fig 4A). We listed the genes as proviral if differential $\geq$ 0 or antiviral if differential $\leq$ 0. **Daniloski et al. [77]:** We used Table S1. FDR MOI1 $\leq$ 0.05 or FDR MOI3 $\leq$ 0.05. **Danziger et al. [11]:** In S1 Table, we used the genes annotated as proviral or antiviral by the authors. **Gordon et al. [76]:** For A549 +ACE2 in Table S6 or Caco2 in Table S7, for proviral genes, we used Averaged z-scores $\leq$ 2, and for antiviral genes, we used Averaged z-scores $\geq$ 2. **Grodzki et al. [75]:** For VeroE6 in additional file 4, tab 4, we used FDR <0.25. For HEK293T +Cas9 Study 1 in additional file 6, tab17, we used FDR <0.25. For HEK293T +Cas9 Study 1 in additional file 7, tab7, we used FDR <0.25. **Hoffmann et al. [28]:** For 37˚C (Table S1E) and for 33˚C (Table S1C), we selected proviral genes if FDR $\leq$ 0.05 and z-score $\geq$ 0 and antiviral genes if FDR $\leq$ 0.05 and z-score $\leq$ 0. **Hossain et al. [69]:** In Figure 3E, we selected the top 15 genes in the spike-mNG axis by negative log robust rank aggregation. **Israeli et al. [74]:** We used Supplementary Data 1. **Kaur et al. [15]:** We used the genes labelled in Fig 1. **Le Pen et al. (this study):** We used z-score $\geq$ 2 for antiviral genes and $\leq$ 2 for proviral genes; see Unbiased arrayed CRISPR KO screen analysis section above for more details. **Loo et al. [73]:** We used the genes labelled in Figure 2. **Mac Kain et al. [18]:** In Electronic Supplementary Material 5, for antiviral genes, we used the filter pos|rank $\leq$13; for proviral genes, we used neg|rank $\leq$13. **Martin-Sancho et al.**

[12]: We used Table S3, "Lentivirus validated hits". **Pahmeier et al. [72]:** We used the genes labelled in Figure 6. **Rebendenne et al. [71]:** For Calu3_Gattinara, we used for proviral genes: residual_z-score_avg $\geq$ 2.5 and for antiviral genes: residual_z-score_avg $\leq$ 2.5 (no gene). For VeroE6, proviral: residual_z-score_avg $\geq$ 2.5, and antiviral: residual_z-score_avg $\leq$ 2.5. For Caco2, proviral: residual_z-score_avg $\geq$ 2.5 and antiviral: residual_z-score_avg $\leq$ 2.5. For Calu3_Calabrese, proviral: residual_z-score_avg $\leq$ 2 and antiviral: residual_z-score_avg $\geq$ 2. **Rehfeld et al. [70]:** In Table S1, we considered genes as hits for PRF-1 top eGFP-mCh or PRF-1 bottom eGFP-mCh if FDR $\leq$ 0.05. **Schneider et al. [31]:** For both 37°C (Table_S1A) and 33°C (Table_S1B), we listed the gene as proviral if FDR $\leq$ 0.05 and z-score $\geq$ 0 and antiviral if FDR $\leq$ 0.05 and z-score $\leq$ 0. **Wang et al. [67]:** We used Table S1. Proviral: Enrichment score $\leq$ 10^(-4). **Wei et al. 2021 [66]:** We used Table S1. For proviral genes, we used Cas9-v1 Avg. $\geq$ 2.5 and Cas9-v2 Avg. $\geq$ 2.5. Average between Cas9-v1 Avg. and Cas9-v2 Avg. is given in the table. For antiviral genes, we used Cas9-v1 Avg. $\leq$ −2.5 and Cas9-v2 Avg. $\leq$ −2.5. Average between Cas9-v1 Avg. and Cas9-v2 Avg. is given in the table. **Wei et al. 2023 [65]:** In Table S1, for Day 7 or Day 14, we used fdr $\leq$ 0.05 for positive regulators of ribosomal frameshifting or negative regulators of ribosomal frameshifting. **Wickenhagen et al. [13]:** We used the genes labelled in Figure 1B. **Xu et al. [19]:** In Huh-7.5 or A549-ACE2 cells, in untreated or in IFN-gamma treatment, we used Log10 $p$-value (mNG-High versus mNG-Low Enrichment) $\geq$ 3. **Zhu et al. [64]:** In Supplementary Data 1, SARS-CoV-2 WT and 2 VOCs tested, for proviral genes, we used pos.score_wt $\leq$ 0.0005 or pos.score_alpha $\leq$ 0.0005 or pos.score_beta $\leq$ 0.0005.

**Human genetic studies.** **Degenhardt et al. [90]:** We used Table 2 and added KANSL1 and TAC4, based on new analysis by Pairo-Castineira et al. 2023 [25]. **Kousathanas et al. [26]:** We used Table 1 from Pairo-Castineira et al. 2023 [25]. **Pairo-Castineira et al. 2021 [98]:** We used Table 1 from Pairo-Castineira et al. 2023 [25]. **Pairo-Castineira et al. 2023 [25]:** We only considered variants near annotated genes (i.e., we excluded rs1073165). **Roberts et al. [83]:** We used Figure 2. **Zhou et al. [95]:** We used Table 1 and the $p$-values from COVID-19 hospitalization (European ancestry only).

**SARS-CoV-2 protein interactomes.** Davies et al. [109], Laurent et al. [102], Li et al. [108], Samavarch-Tehrani et al. [100], St-Germain et al. [103], Stukalov et al. [107]: We used the Supplementary Table 3 from [27]. Gordon et al. [101]: We used the genes listed in Table S2. Liu et al. [104] and May et al. [105]: We used the genes listed in May et al. [105] Table S5-new. Zhou et al. [106]: We used the genes listed in Table S1. "SARS-CoV-2-human protein-protein interactions identified in this study."

**SARS-CoV-2 RNA interactomes.** Flynn et al. [114], Kamel et al. [113], Labeau et al. [112], Lee et al. [115], Schmidt et al. 2021 [111]: We used the Supplementary Table 3 from [27]. Schmidt et al. 2023 [110]: We used Table S2, "Huh-7 interactome comparison tab", genes listed in the following categories: "Huh-7 gRNA FDR5 HS" and "Huh-7 sgmRNA FDR5 HS".

**Altered phosphorylation states in SARS-CoV-2-infected cells.** **Bouhaddou et al. [116]:** For Vero E6 cells, we used Table S1, tab 1 "PhosphoDataFull" and filtered for adj.pvalue $\leq$ 0.05 and log2FC $\geq$ 1 or adj.pvalue $\leq$ 0.05 and log2FC $\leq$ −1 in at least 3 different time points.

## Generation of PLSCR1 KO cells

**CRISPR KO.** KO Huh-7.5 and A549-ACE2 cells were generated using 2 anti-PLSCR1 Edit-R crRNAs from Horizon Discovery (cat. CM-003729-02-0002 and CM-003729-04-0002) or nontargeting controls (cat. U-007501-01-05 and U-007502-01-05) resuspended with an equimolar amount of Edit-R tracrRNA (Horizon, cat. U-002005-20) to form sgRNAs. The sgRNAs were then cotransfected with Cas9-mKate2 mRNA (Horizon, cat. CAS12218) according to the manufacturer's protocol. Cells were examined for mKate2 signal and FACS sorted into bulk

and single cell populations, gating on mKate2 signal, 24 to 48 hours after transfection. Bulk and single-cell populations were then assessed for PLSCR1 expression by WB to confirm KO.

**Amplicon sequencing.** Genomic DNA was isolated from a frozen cell pellet using the Qiagen DNeasy kit (Qiagen, cat. 69504) and treated with RNAse A in the optional RNA digestion step. The region of interest was then amplified using Q5 2× mastermix (NEB, cat. M0492S), 500 ng of template DNA, 0.5 μM of forward and reverse primers, and the following PCR conditions: 98°C for 30 seconds, followed by 98°C for 5 seconds, 64°C for 15 seconds, and 72°C for 20 seconds, repeating those steps 30 times before holding at 72°C for 2 minutes. The primers used when amplifying PLSCR1 genomic DNA from WT and KO Huh-7.5 and A549+ACE2 cells were RU-O-32687 (5′ AACATAGAGGTGATTATGATTTCGTCT) and RU-O-32526 (5′ GGAGGAGCTTGGATTTCTATCTAC). PCR reactions were run on a 1% agarose gel to confirm amplification. Amplicons were purified with a Zymo DNA clean and concentrator kit (Zymo, cat. D4013) before sending to Genewiz for amplicon sequencing.

**Western blots.** Cell pellets were collected and lysed in RIPA buffer (Thermo, catalog number 89900) with 1× Halt protease inhibitor cocktail and 1× EDTA (Thermo, catalog number 87786). Cell lysates were spun down in a refrigerated centrifuge at 15,000 x g at 4°C for 15 minutes to pellet any cell debris, and the supernatant was collected and transferred to another tube. The collected samples were quantified by BCA assay (Thermo Scientific, cat. #23225). Before loading into the gel, we added sample buffer (Thermo, catalog number NP0007) with β-mercaptoethanol and heated the sample at 95°C for 10 minutes. Samples were allowed to cool back to room temperature before loading into 12% Bis-Tris 1.0 mm gels (Invitrogen, cat. #NP0321BOX). Proteins were electrophoretically transferred onto nitrocellulose membranes. Membranes were blocked with 5% fat-free milk in 1× TBS (Thermo, catalog number NP00061) and then incubated with primary antibody at 4°C overnight in 5% fat-free milk in 1× TBS with 0.5% Tween-20 (TBST). Primary antibody: rabbit anti-PLSCR1 polyclonal antibody (Proteintech, cat. #11582-1-AP) and mouse anti-β-actin antibody (Millipore Sigma, cat. A5316-100UL) as a loading control. After incubation, membranes were washed 3 times with 1× TBST and then incubated with fluorescently conjugated secondary antibodies for 2 hours at room temperature. Secondary antibodies: LI-COR IRDye goat anti-rabbit 800 and goat anti-mouse 680 (LI-COR cat. 926-32211 and 926-68070, respectively). Membranes were washed 3 times with 1× TBST, once with 1× TBS, then imaged on an Azure 600. For the WB in Supplementary Figure 4, this protocol was modified slightly: Proteins were electrophoretically transferred onto 0.22 μm polyvinylidene difluoride (PVDF) membranes, incubated with a primary antibody solution of rabbit anti-PLSCR1 polyclonal antibody (Proteintech, cat. #11582-1-AP) and polyclonal rabbit anti-RPS11 antibody (Abcam, cat. ab157101), a secondary antibody solution of goat anti-rabbit HRP (Invitrogen, cat. 31462) and visualized using a SuperSignal West Femto Maximum Sensitivity Substate kit (Thermo, cat. #34096).

## Cell viability assay

On day 0, 4,000 A549-ACE2 cells/well or 8,000 Huh-7.5 cells/well were seeded in 100 μL media (DMEM, 10% FBS, 1× NEAA, 1× Penicillin-Streptomycin) in a 96-well plate. The next day, blasticidin selection was added as indicated in the figure to serve as a control for reduced cell viability. On day 4, cell viability was assessed by resazurin assay (Abcam, cat. ab129732) according to the manufacturer's protocol.

## JAK-STAT inhibitor treatment

InSolution (Millipore, cat. 420097-500UG) was used according to the manufacturer's instructions.

## Titration of IFN-a2a in CHIKV-infected cells

Approximately 6,000 Huh-7.5 cells/well were seeded in 100 μL media (DMEM, 10% FBS, 1X NEAA). The following day, we treated cells with 1 of 12 concentrations of IFN-a2a (PBL, cat. 11101-2): 64 pM, 32 pM, 16 pM, 8 pM, 4 pM, 2 pM, 1 pM, 0.5 pM, 0.25 pM, 0.125 pM, 0.0625 pM, and 0 pM. The following day, the cells were infected with 2 μL of CHIKV-181/25-mKate2 (approximately 17,000 FFU per well, titer determined on Huh-7.5 cells) and fixed after 12 hours. Plates were stained with a 1:1,000 dilution of Hoechst for at least 10 minutes before washing with PBS and imaging for mKate2 signal.

## RT-qPCRs on ISGs

Huh-7.5 and A549-ACE2 cells were seeded at densities of 36,000 or 18,000 cells/well, respectively, in 500 μL of media (DMEM, 10% FBS, 1× NEAA) in 24-well plates. The following day, a dilution series of IFN-α2a (PBL, cat. 11101-2) or IFN-β (PBL, cat. #11415) was prepared (64 pM, 32 pM, 16 pM, 8 pM, 4 pM, 2 pM, 1 pM, 0.5 pM, 0.25 pM, 0.125 pM, 0.0625 pM, and 0 pM) and 50 μL of each dilution added to the cells in duplicate. After 24 hours, the media was removed and the cells were washed with 1 mL of ice-cold PBS. RNA Lysis Buffer (200 μL, Zymo Research, cat. #R1060-1-100) was added to the cells, and the plates were frozen at−20˚C before RNA isolation.

RNA was extracted using the Zymo Quick RNA 96-kit (Zymo Research, cat. R1052) including DNaseI treatment, followed by cDNA synthesis using the SuperScript IV VILO Master Mix (Invitrogen, cat. 11756050) according to manufacturers' instructions. qPCRs were conducted on a QuantStudio 3 cycler using the Taqman Fast Advance master mix (Life Technologies Corporation, cat. 4444965) and the following assays: *RPS11* (ThermoFisher 4331182; Hs01574200_gH), *IFI6* (ThermoFisher 4331182; Hs00242571_m1), *OAS1* (ThermoFisher 4331182; Hs00973635_m1). *IFI6* and *OAS1* were normalized to *RPS11* mRNA levels using the deltaCt method [194].

## PLSCR1 subcellular localization

**IF staining.** A549-ACE2 cells were plated onto #1.5, 12 mm glass coverslips (Fisher Scientific, cat. #1254581) placed at the bottom of the wells of a 24-well plate. When confluent, the cells were fixed, permeabilized with 1% Triton X-100 for 5 minutes, and blocked for 1 hour at room temperature with 1 mL of PBS-BGT (1× PBS with 0.5% bovine serum albumin, 0.1% glycine, 0.05% Tween 20). Afterward, the cells were incubated in a 1:500 dilution of 4D2 mouse anti-PLSCR1 antibody (Millipore Sigma, cat. #MABS483) in PBS-BG (1× PBS with 0.5% bovine serum albumin and 0.1% glycine) overnight at 4˚C with rocking. The cells were then washed twice with PBS-BGT before incubation with a secondary antibody solution of 1:1,000 anti-mouse 588 (Thermo Fisher, cat. #A-11001) and 1:1,000 Hoechst dye (Thermo Fisher, cat. #62249) in PBS-BG for 2 hours at room temperature, followed by 3 washes with PBS-BGT. For those images where cells were treated with IFN shown in **S11B Fig**, A549-ACE2 cells were treated with 20 pM of IFN-α2a (PBL, cat. 11101-2) for 48 hours before harvest, then fixed and stained as described above. For those images where cells were infected with SARS-CoV-2 shown in **S11C Fig**, A549-ACE2 cells were infected with approximately 3,400,000 PFU of SARS-CoV-2 (titer determined on Vero E6 cells) diluted in 200 μL Opti-MEM to achieve almost 100% infection. After 24 hours, cells were fixed and stained as described above, with slight modification to the antibody solutions. Primary antibody solution: 1:500 4D2 mouse anti-PLSCR1 and 1:3,000 rabbit anti-SARS-CoV-2 N (Genetex, cat. GTX135357) in PBS-BG. Secondary antibody solution: 1:1,000 goat anti-mouse 594, 1:1,000 goat anti-rabbit 647, and 1:1,000 Hoechst dye.

**Imaging.**   The coverslips were mounted onto slides (Fisher Scientific, cat. #1255015) with Invitrogen ProLong Gold Antifade Mountant (Fisher Scientific, cat. # P36930). The slides were allowed to cure for 24 hours before the edges of the coverslips were sealed. For **Fig 5A**, the cells were imaged by confocal microscopy. Confocal images were acquired using Zeiss Zen Blue (v3.5) software on a LSM 980 point scanning confocal microscope (Zeiss) hooked to a Axio Observer.Z1 / 7 stand equipped with C Plan-Apochromat 63×/1.40 oil (RI:1.518 at 23˚C) objective lens (Zeiss). CW excitation laser lines 405 nm and 488 nm were used to excite the fluorescence of DAPI and AF488 labeled samples. Emitted fluorescence were spectrally grated (410 to 483 nm for DAPI, 499 to 552 nm for AF488) to avoid fluorescence bleed through and were detected in MA-PMT (DAPI) and GaAsP-PMT (AF488). The confocal pinhole was set to 1AU for AF488, and the detector master gains were set within the linear range of detection (550 to 750 V). For the IFN-treated and SARS-CoV-2-infected cells shown in **S11B and S11C Fig**, these were mounted as described previously but imaged via widefield imaging. Widefield imaging was performed in the Rockefeller University's Bio-Imaging Resource Center, RRID: SCR_017791, on a DeltaVision Image Restoration Microscope (Applied Precision, now Leica Microsystems) using inverted IX-70 stand (Olympus, Evident Scientific) and a 60× NA 1.42 oil immersion objective (Olympus), and filter sets for DAPI (Ex: 390 ± 18, Em: 435 ± 48), Alexa Fluor 594 (Ex: 575 ± 25, Em: 632 ± 60), Alexa Fluor 647 (Ex: 575 ± 25, Em: 676 ± 34). Images were acquired on a pco.edge sCMOS camera and were subsequently deconvolved using the Deltavision image acquisition software (SoftWorx). Scanned images for both confocal and widefield imaging were saved as .czi files.

## Focus-forming assay on SARS-CoV-2-infected cells

In **Fig 5B**, Huh-7.5 and A549-ACE2 cells were cultured in media (DMEM with 5% FBS) and seeded at densities of $2 \times 10^5$ and $1 \times 10^5$ cells per well, respectively, in collagen-coated 12-well plates to reach 80% to 90% confluency by the day of infection. A 1:10 serial dilution of virus stock was made in Opti-MEM in 5 separate tubes. Media was aspirated from the cells, and the wells were washed with 1 ml of PBS before adding 200 μL of each virus dilution to the cells in triplicate. Plates were incubated at 37˚C with 5% $CO_2$ for 1 hour, rocking every 15 minutes for even virus distribution. A 1% methylcellulose overlay medium was prepared and mixed with complete growth media at 37˚C; 2 ml of this overlay was added to each well after removing the virus inoculum. Plates were then incubated at 37˚C with 5% $CO_2$ for 48 hours for Huh-7.5 cells or 72 hours for A549-ACE2 cells. Cells were then fixed in final 10% neutral buffered formalin and IF stained as described in the Unbiased arrayed CRISPR screen section. PLSCR1 KO and WT cells were compared at similar virus dilutions.

In **Fig 6G**, the above protocol was followed to titer SARS-CoV-2 strains on Huh-7.5 PLSCR1 KO cells. Then, Huh-7.5 WT and KO cells were seeded at $2 \times 10^5$ cells per well in 1 mL of media (DMEM, 10% FBS, 1× NEAA) in 12-well plates to reach 80% to 90% confluency the next day. Media was aspirated from the cells, the wells were washed with 1 mL of PBS, and then the cells were infected with approximately 50 FFU of SARS-CoV-2 (for each strain) as determined on PLSCR1 KO Huh-7.5 cells diluted in 200 μL of Opti-MEM. Plates were then incubated, overlayed with methylcellulose, fixed, and stained as described above.

## Transfection with SARS-CoV-2 replicon system

The SARS-CoV-2 replicon and the method for electroporation has been described previously [119]. Briefly, $6 \times 10^6$ Huh-7.5 WT and PLSCR1 KO cells were electroporated at 710 V with 2 μg of SARS-CoV-2 N mRNA and 5 μg of replicon RNA. The cells rested for 10 minutes at room temperature before resuspending to a concentration of 300,000 cells/mL and plating

100 μL of cells into each well of a 96-well plate. After 24 hours, supernatant was collected from the replicon-transfected cells and assayed for *Renilla* luciferase activity according to kit instructions (Promega, cat. E2810).

## Transduction with SARS-CoV-2 spike/VSV-G-pseudotyped, single-cycle, replication-defective HIV-1 viruses

**Virus preparation.** SARS-CoV-2 spike/VSV-G-pseudotyped, single-cycle, replication-defective HIV-1 viruses (pCCNanoLuc/GFP) were prepared as in [120]. Plasmids were a kind gift of Theodora Hatziioannou and Paul D. Bieniasz (The Rockefeller University, New York, USA) [120,195,196]. One day before the transfection, $4 \times 10^6$ 293T cells were seeded in a 10-cm dish. One hour prior to transfection, the growth media in the dish was replaced with 9 mL of fresh media containing 2% serum. A 1,000-μL transfection mixture was prepared, comprising the diluent (a 150-mM NaCl solution prepared with sterile cell culture water), 5 μg of HIV GP plasmid, 5 μg of pCLG plasmid, and either 2.5 μg of SARS-CoV-2 spike Δ19 or 1 μg of pHCMV.G plasmid, ensuring the total plasmid content did not exceed 12.5 μg. After brief vortexing, 50 μL of PEI (1 mg/mL, Polysciences cat. 23966) was added to achieve a 1:4 DNA/PEI ratio. The mixture was vortexed for 5 seconds and then allowed to sit for 20 minutes in a hooded environment. Following gentle mixing by pipetting, 1 mL of the transfection mixture was added to the 10-cm dish. Media was changed 12 hours posttransfection, and the supernatant was harvested and filtered through a 0.2-μm filter 48 hours posttransfection, then stored at −80˚C.

**Transduction of PLSCR1 KO or WT A549-ACE2 cells.** Seeded in 96-well plates, 6,000 A549-ACE2 cells per well were cultured in 100 μL of media. After 2 days, either 10 μL of SARS-CoV-2 spike pseudotyped virus or 0.01 μL of VSV-G pseudotyped virus were diluted in a final volume of 100 μL of media and added to the wells to yield comparable NanoLuc signals. Plates were then spun at 200*g* for 5 minutes and incubated at 37˚C. Two days postinfection, the media was aspirated and replaced with 50 μL of NanoGlo solution, sourced from the Promega Nano-Glo Luciferase Assay kit (Promega, N1110), with a substrate to buffer ratio of 1:100. NanoLuc signal was subsequently quantified using a Fluostar Omega plate reader.

**Transduction of siRNA-treated HEK293T cells.** Seeded in 96-well plates, 1,600 HEK293T, HEK293T-ACE2, or HEK293T-ACE2-TMPRSS2 cells were cultured in 80 μL of media. The next day, a 20-μL transfection mixture made of Opti-MEM, 1% DharmaFECT1 (Horizon, T-2001-03), and 250 nM siRNA, PLSCR1 ON-TARGETplus SMARTpool siRNA (Horizon, cat. L-003729-00-0005) or nontargeting control (Horizon, cat. D-001810-10-05) was added to the cells. The final concentration of siRNAs was 25 nM. After 2 days, either 2 μL of SARS-CoV-2 spike pseudotyped virus or 0.2 μL of VSV-G pseudotyped virus were diluted in a final volume of 100 μL of media and added to the wells to yield comparable NanoLuc signals. Plates were then spun at 200*g* for 5 minutes and incubated at 37˚C. Two days after infection, NanoLuc signal was quantified as described in the "Infection of PLSCR1 KO or WT cells" section.

## Infection of siRNA-treated A549 cells with SARS-CoV-2

Seeded in 96-well plates, 1,000 A549, A549-ACE2, or A549-ACE2-TMPRSS2 cells were cultured in 80 μL of media. On the same day, a 20-μL transfection mixture made of Opti-MEM, 1% DharmaFECT1 (Horizon, T-2001-03), and 250 nM siRNA, PLSCR1 ON-TARGETplus SMARTpool siRNA (Horizon, cat. L-003729-00-0005) or nontargeting control (Horizon, cat. D-001810-10-05) was added to the cells. The final concentration of siRNAs was 25 nM. Three days after transfection, the cells were infected by adding 34,000 PFU of SARS-CoV-2 (titer

determined on Vero E6 cells) diluted in 10 μL media to each well. Plates were then spun at 200$g$ for 5 minutes and incubated at 37°C. Staining and readout as described above in the "Unbiased arrayed CRISPR KO screening" section.

## Pan-virus infection of PLSCR1 KO cells

A549-ACE2 cells were seeded at a density of 6,000 cells/well in 96-well plates in 90 μL media. The following day, 10 μL diluted virus was added to each well. Virus concentrations are as follows: CHIKV-mKate, 0.05 μL virus stock per well (titer $8.5 \times 10^6$ PFU/mL determined in Huh-7.5 cells); hCoV-NL63, 10 μL virus stock per well (titer $1.4 \times 10^5$ PFU/mL); hCoV-OC43, 10 μL virus stock per well (titer $1.06 \times 10^7$ PFU/mL); hPIV-GFP, 0.05 μL virus stock per well; HSV1-GFP, 0.5 μL virus stock per well (titer $2.4 \times 10^8$ PFU/mL determined on Vero E6 cells); IAV WSN, 0.5 μL virus stock per well; SARS-CoV-2, 0.5 μL virus stock per well (titer $3.4 \times 10^6$ PFU/mL determined on Vero E6 cells); SINV-Toto1101-mScarletI, 10 μL virus stock per well; VEEV-EGFP, 0.005 μL virus stock per well (titer $1.45 \times 10^9$ PFU/mL determined on BHK-21); VSV-GFP, 0.05 μL virus stock per well; YFV_17D, 5 μL virus stock per well. The cells were fixed, stained, and imaged as described in the "Unbiased arrayed CRISPR KO screen" section. Fluorescent viruses were not stained: The fluorescent signal was used as a reporter. We used the following primary antibodies when applicable: anti-dsRNA (J2) mouse (Nordic MUbio, cat. 10010200) diluted 1:500 was used for hCoV-NL63 and hCo-V-OC43, anti-IAV mouse (Millipore, cat. MAB8257) diluted 1:3,000, anti-YFV mouse (Santa Cruz Biotechnology, cat# sc-58083) diluted 1:500, anti-SARS2-S rabbit (Genetex, cat. GTX135357) diluted 1:3,000.

Huh-7.5 cells were transfected with a 1:200 dilution of Dharmafect 4 (Horizon, cat. T-2004-01) and 25 nM ON-TARGETplus SMARTpool siRNAs (Horizon Discovery) in 96-well plates. The cells were infected 3 days after siRNA transfection with hCoV-NL63 or hCoV-OC43 or 4 days after siRNA transfection with SARS-CoV-2 or hCoV-229E. IF and imaging as described above.

## Measuring extracellular ACE2 expression using flow cytometry

A549-ACE2 cells were detached from the wells using Accutase (Innovative Cell Technologies, Cat: AT-104) for 5 minutes at 37°C. Following detachment, cells were rinsed once with PBS and stained with LIVE/DEAD Fixable Aqua (Invitrogen, Cat: L34966, diluted 1:1,000 in PBS) for 15 minutes at room temperature in the dark. After staining, cells were rinsed once more with PBS. Then, cells were incubated with FcR blocking reagent (Miltenyi Biotec, 1:50) and ACE2-AF488 (Sino Biological, Cat: 10108-MM37-F-100, 1:30) in FACS buffer (2% FBS in PBS) for 30 minutes at 4°C in the dark. Following incubation, cells were washed twice with FACS buffer and subjected to flow cytometry analysis. Samples were acquired on an LSRII flow cytometer (BD Biosciences), and the results were analyzed with FlowJo software (Tree Star). Before analyzing cell surface ACE2 expression, gating procedures were implemented to exclude dead cells and doublets from the analysis.

## SARS-CoV-2 variant infections

For Fig 6A–6E, Huh-7.5 cells were plated at 7,500 cells/well in 100 μL of media (DMEM, 10% FBS, 1× NEAA) in 96-well plates. The next day, cells were infected with quantities of virus that yielded comparable percent infections in the WT cells, then spun at 200$g$ for 5 minutes and incubated at 33°C. The quantities of virus used were as follows: 0.1 μL/well for parental, 0.05 μL/well for beta, 1 μL/well for delta, 0.5 μL/well for Omicron BA.5, and 0.05 μL/well for Omicron XBB.1.5. The infected cells were fixed after 24 hours and stained as described previously.

## Growth curves using the parental SARS-CoV-2 strain and Omicron (XBB.1.5)

Huh-7.5 WT and PLSCR1 KO cells were plated at 100,000 cells/well in 1 mL media (DMEM, 10% FBS, 1× NEAA) in a 24-well plate. The following day, cells were infected at MOI 0.01 (as titered on PLSCR1 KO Huh-7.5 cells) with the New York 2020 isolate of SARS-CoV-2 ("parental") and SARS-CoV-2 Omicron XBB.1.5. Plates were incubated for 1 hour at 37˚C. Inoculum was removed and saved as the 0-hour time point. Cells were washed twice with PBS. Fresh media was added and then collected as the 1-hour time point. Fresh media was added again, and the cells were incubated at 33˚C. All supernatant was collected from the cells again at 24 hours, 48 hours, and 72 hours and was replaced with fresh media. Collected supernatant was frozen at −80˚C until titering. Supernatants were titered by plaque assay as follows: Vero E6 cells were plated at 200,000 cells/well in 12-well plates. The following day, collected supernatant was 10-fold serially diluted, the media removed from the plates, and 300 μL diluted virus added. Each sample of supernatant was titered in duplicate. Cells were incubated with virus for 1 hour at 37˚C, then inoculum was removed and 1 mL of 1% methylcellulose (supplemented with 5% FBS). Plates were incubated for 3 days at 37˚C, then fixed with 2 mL of 7% formaldehyde and stained with crystal violet before being rinsed with water, left to dry overnight, and the plaques counted.

**Generating and infecting cells overexpressing ISGs (including PLSCR1).** **Plasmid sources.** WT PLSCR1, IFITM2, and LY6E overexpression lentiviral plasmids originated from the SCRPSY library [134]. CH25H originated from the Thermo Scientific Precision LentiORF Collection. RRL.sin.cPPT.CMV/Flag-NCOA7 variant 6.IRES-puro.WPRE (CG494) was a gift from Caroline Goujon (Addgene plasmid # 139447) [160].

**Plasmid cloning.** N-terminal 3x FLAG-tagged PLSCR1, C-terminal 3x FLAG-tagged PLSCR1, and PLSCR1 H262Y in the SCRPSY vector, used in **Figs 4B and 7D**, were generated by designing and ordering large dsDNA gene blocks of PLSCR1 that contained the desired mutations from IDT. These gene blocks were cloned into the PLSCR1-SCRPSY vector [134] and confirmed by sequencing using Plasmidsaurus (see **S18 Table** for sequences).

PLSCR1 variants of interest (**Figs 7C and S12**) were cloned by site directed mutagenesis using the NEB Quick Protocol for Q5 Site-Directed Mutagenesis Kit (cat. E0552) according to the manufacturer's instructions. For G20V (rs79551579), we used the primers JLP621-O-01/02; for A29T (rs41267859), we used the primers JLP621-O-03/04; for I110fs rs749938276, we used the primers JLP621-O-07/08; and for H262Y (rs343320), we used the primers JLP621-O-09/10. Primer sequences are listed in **S20 Table**.

**Lentivirus production.** Lentivirus were generated in Lenti-X 293T cells by transfecting 200 ng VSV-G plasmid, 700 ng Gag-Pol plasmid, and 1,100 ng plasmid of interest with lipofectamine 2000 in DMEM supplemented with 5% FBS. Media was changed 4 to 6 hours later, and lentivirus harvested at 24 and 48 hours. Lentivirus-containing supernatants from both time points were pooled, then filtered through a 0.45-μM filter before aliquoting into 2 mL tubes and freezing at −80˚C until use.

**Cell transduction.** In a 12-well plate, 0.3 million cells were transduced in suspension using a serial dilution of each lentivirus in DMEM (Fisher Scientific, cat. #11995065) supplemented with 5% FBS (HyClone Laboratories, Lot. #KTH31760) and 4 μg/mL polybrene (Millipore, cat. #TR-1003-G). The cells were then spinoculated at 37˚C for 1 hour at 1,000 x *g*. The following day, the cells were split into two 6-well plates. After 24 hours, one of the duplicates was treated with 2 μg/mL puromycin to select for transduced cells. Further experiments were carried out using the cells that had approximately 30% transduction before selection.

**SARS-CoV-2 infection.** In **Fig 7D**, Huh-7.5 and A549+ACE2 cells were plated at 6,000 cells/well and 3,000 cells/well, respectively, in 100 μL of media (DMEM, 10% FBS, 1× NEAA)

in 96-well plates. The following day, cells were treated with IFN (10 pM for Huh-7.5 cells and 20 pM for A549+ACE2 cells). On the third day, the Huh-7.5 cells were infected by adding 10 μL of media containing 0.1 μL of virus (titer of $3.4 \times 106$ PFU/ml, as measured on Vero E6 cells) to each well, and the A549+ACE2 cells were infected by adding 10 μL of media containing 1 μL of virus to each well. After adding the virus, the plates were spun at 200$g$ for 5 minutes and incubated at 37˚C. Plates were harvested the next day by fixing and staining as described above.

In **Figs 7C, S8A, and S12**, A549+ACE2 cells were plated at 7,000 cells/well in 100 μL of media (DMEM, 10% FBS, 1× NEAA) in 96-well plates. The following day, cells were infected by adding 10 μL of media containing 1 μL of virus (titer of $3.4 \times 106$ PFU/ml, as measured on Vero E6 cells) to each well as described above.

**IF staining and imaging.** In **Figs 7C-7E, S8A, and S12**, cells were stained for IF as described in the "Unbiased arrayed CRISPR KO screening" section, with different primary and secondary antibody solutions. The primary antibody solution was a 1:3,000 dilution of rabbit anti-SARS-CoV-2 nucleocapsid polyclonal antibody (Genetex, cat. #GTX135357) and a 1:500 dilution of 4D2 mouse anti-PLSCR1 antibody (Millipore Sigma, cat. #MABS483) in PBS, and the secondary antibody solution was 1:1,000 goat anti-rabbit 594 (Thermo Fisher, cat. #A-11012) or 1:1,000 goat anti-rabbit Alexa Fluorv 647 (Thermo Fisher, cat. #A-21245), 1:1,000 goat anti-mouse Alexa Fluor 488 (Thermo Fisher, cat. #A-11001), and 1:1,000 Hoechst dye (Thermo Fisher, cat. #62249) in PBST.

## SARS-CoV-2 infection of human SV40-fibroblasts-ACE2

**Patients.** Written informed consent was obtained in the country of residence of the patients, in accordance with local regulations, and with institutional review board (IRB) approval. Experiments were conducted in the US in accordance with local regulations and with the approval of the IRB. Approval was obtained from the Rockefeller University Institutional Review Board in New York, USA (protocol no. JCA-0700).

PLSCR1 H262Y was genotyped using the primers JLP616-O-1-4 listed in **S20 Table**.

**Generation of human SV40-fibroblasts ACE2 stable cell lines.** *ACE2* cDNA was inserted with In-Fusion cloning kit (Takara Bio) and using the XhoI and BamHI restriction sites into linearized pTRIP-SFFV-CD271-P2A in accordance with the manufacturers' instructions. We checked the entire sequence of the *ACE2* cDNA in the plasmid by Sanger sequencing. Then, HEK293T cells were dispensed into a 6-well plate at a density of $8 \times 10^5$ cells per well. On the next day, cells were transfected with pCMV-VSV-G (0.2 μg), pHXB2-env (0.2 μg; NIH-AIDS Reagent Program; 1069), psPAX2 (1 μg; Addgene plasmid no. 12260), and pTRIP-SFFV-CD271-P2A-ACE2 (1.6 μg) in Opti-MEM (Gibco; 300 μL) containing X-treme-Gene-9 (Sigma Aldrich; 10 μL) according to the manufacturers' protocol. After transfection for 6 hours, the medium was replaced with 3 mL fresh culture medium, and the cells were incubated for a further 24 hours for the production of lentiviral particles. The viral supernatant was collected and passed through a syringe filter with 0.2 μm pores (Pall) to remove debris. Protamine sulfate (Sigma; 10 μg/mL) was added to the supernatant, which was then used immediately or stored at −80˚C until use.

For the transduction of SV40-fibroblasts with ACE2, $5 \times 10^5$ cells per well were seeded in 6-well plates. Viral supernatant was added (500 μL per well). The cells were then further incubated for 48 hours at 37˚C. Cells were keep in culture and after 8 days, transduction efficiency was evaluated by CD271 surface staining with CD271 AlexaFluor 647, 1:200 dilution (BD Pharmigen 560326). MACS-sorting was performed with CD271-positive selection beads (Miltenyi Biotec) if the proportion of CD271-positive cells was below 80% [8,178].

**Infection.**   In a 96-well plate, 5,000 cells per well were seeded and infected the next day with SARS-CoV-2 at MOI = 0.05, using a titer determined on Vero E6 cells. Cells were fixed at 2 dpi, stained, and imaged as described in the "Unbiased arrayed CRISPR KO screen" section.

**Dryad DOI.**   Available on Dryad (DOI: 10.5061/dryad.6q573n65k).

## Supporting information

**S1 Fig.** (**A**) A schematic of SARS-CoV-2 entry and the sites where known ISG entry restriction factors function is shown; adapted from [1]. CD74 suppresses endolysosomal cathepsins, enzymes that process certain viral glycoproteins to make them fusion-competent [148,198]. CH25H facilitates the sequestration of accessible cholesterol, which results in decreased virus–cell membrane fusion and viral entry [14,63]. NCOA7 accelerates the acidification of the lysosome, leading to the degradation of viral antigens [160,161]. The role of IFITM1-3 proteins in SARS-CoV-2 entry has been the subject of numerous studies, which have sometimes yielded varying conclusions [1,122,199–205]. However, it appears that (i) IFITM2 restricts WT SARS-CoV-2 endosomal entry [122], (ii) recent SARS-CoV-2 variants may evade IFITM2 [199], and (iii) IFITM3 KO mice are hypersusceptible to WT SARS-CoV-2 [206]. LY6E and PLSCR1 restrict virus–cell membrane fusion at the endosome through unknown mechanisms; see [19,121] and this study. (**B**) TMPRSS2 WB in Huh-7.5-Cas9 cells, both WT and TMPRSS2 KO, conducted as in the whole-genome arrayed screen (**Fig 1**), and in A549-ACE2 (A549 + A) and A549-ACE2-TMPRSS2 (A549 + A/T) cells. ISG, IFN-stimulated gene; KO, knockout; LY6E, lymphocyte antigen 6 family member E; NCOA7, nuclear receptor coactivator 7; PLSCR1, phospholipid scramblase 1; SARS-CoV-2, Severe Acute Respiratory Syndrome Coronavirus 2; TMPRSS2, transmembrane serine protease 2; WB, western blot; WT, wild type. (EPS)

**S2 Fig.** (**A**) Model of protein levels of the gene product over time after complete KO, considering the Huh-7.5 cell division rate (1.7 times per day) and protein half-lives in hepatocytes from [207]. (**B**) Nuclei count (z-score) in arrayed genetic screen without SARS-CoV-2 infection. Examples of VCP (essential gene) and PLSCR1 (SARS-CoV-2 antiviral hit) are plotted. (**C**) Volcano plot of Huh-7.5 cells mRNA-seq treated with 0.5 nM IFN-α2a for 24 hours as in **Fig 1B**. (**D**) GSEA conducted on the arrayed CRISPR KO screens data represented in **Figs 1 and 2**. Description of the top pathways ranked by $p$-value for each quadrant. Databases: [1]Reactome; [2]WikiPathways; [3]Pathway Interaction Database; [4]KEGG; [5]Biocarta. The data underlying this Figure can be found in **S1 Table**. GSEA, gene set enrichment analysis; KO, knockout; PLSCR1, phospholipid scramblase 1; SARS-CoV-2, Severe Acute Respiratory Syndrome Coronavirus 2. (EPS)

**S3 Fig.** (**A**) Occurrence of human genes interacting with SARS-CoV-2 drawn from a selection of 67 large-scale studies. The occurrence reflects the number of independent studies finding each gene as significant. (**B**) Upset plot on data as in (**A**), showing the overlap in significant genes in large-scale SARS-CoV-2 studies by category. (EPS)

**S4 Fig. Cross comparison between proviral and antiviral hits reported in this study and 15 published whole-genome pooled screens [19,31,64–67,69–71,74,75,77–80].** *P* value: Fisher's exact test. (EPS)

**S5 Fig.** (**A**) WB on PLSCR1 KO cells against PLSCR1 (green) and ß-actin (red). (**B**) Cells as indicated were seeded at similar density, treated or not with Blasticidin (used here as a control to decrease cell viability), and cultured for 4 days before resazurin cell viability assay. $n = 4$ independent wells. (**C**) WB on PLSCR1 WT and KO A549-ACE2 cells against PLSCR1 (green) and ß-actin (red). (**D**) Single-cell RNA-seq data from [118], showing the *PLSCR1* mRNA expression in some SARS-CoV-2 target cells. The data underlying this Figure can be found in **S1 Table**. KO, knockout; PLSCR1, phospholipid scramblase 1; SARS-CoV-2, Severe Acute Respiratory Syndrome Coronavirus 2; WB, western blot; WT, wild type.
(EPS)

**S6 Fig.** (**A**) Huh-7.5 WT and PLSCR1 KO cells were infected with 17,000 FFU of CHIKV 181/25 mKate2 for 12 or 24 hours, then fixed, IF stained for nuclei, and the percentage of positive cells determined by imaging for mKate2 reporter signal. $n = 12$ separate wells infected on the same day. Error bars represent SD. ns, nonsignificant; two-way ANOVA. (**B**) Huh-7.5 cells, PLSCR1 KO as indicated, were pretreated with different amounts of IFN-α2a, then infected with 17,000 FFU of CHIKV-mKate for 12 hours; $n = 7$ separate wells infected on the same day; error bars represent SD. (**C-J**) Cells were treated for 24 hours by IFN, as indicated, followed by RT-qPCRs on ISGs. The data underlying this Figure can be found in **S1 Table**. CHIKV, chikungunya virus; FFU, focus-forming unit; IF, immunofluorescence; ISG, IFN-stimulated gene; KO, knockout; PLSCR1, phospholipid scramblase 1; RT-qPCR, quantitative reverse transcription PCR; WT, wild type.
(EPS)

**S7 Fig.** (**A**) WB against PLSCR1 and RPS11. Cas-9-expressing Huh-7.5 cells were transfected with 4 gRNA pools targeting PLSCR1 or nontemplate control (as indicated), and cells were in culture for 7 days. (**B**) Quantification of bands intensity in (**A**). (**C**) mRNA-seq on PLSCR1 KO Huh-7.5 cells as in (**A, B**). The data underlying this Figure can be found in **S1 Table**. KO, knockout; PLSCR1, phospholipid scramblase 1; WB, western blot.
(EPS)

**S8 Fig.** (**A**) A549-ACE2 PLSCR1 KO cells were transduced to stably and ectopically express the indicated genes. The cells were then infected for 24 hours with parental SARS-CoV-2. SARS-CoV-2 N was stained by IF and the percentage of positive cells determined by imaging. $n = 4$ separate wells infected on the same day. Error bars represent SD; **, $p \leq 0.01$; ****, $p < 0.0001$; one-way ANOVA. (**B**) Normalized reads count from mRNA-seq on A549 cells treated or not with 0.5 nM IFN-α2a for 24 hours. (**C**) Normalized reads count from mRNA-seq on Huh-7.5 cells treated or not with 0.5 nM IFN-α2a for 24 hours. The data underlying this Figure can be found in **S1 Table**. IF, immunofluorescence; KO, knockout; PLSCR1, phospholipid scramblase 1; SARS-CoV-2, Severe Acute Respiratory Syndrome Coronavirus 2.
(EPS)

**S9 Fig.** (**A-I**) Infection of A549-ACE2 cells with viruses as indicated. IF staining was used for IAV WSN, hCoV-OC43, and SARS-CoV-2, otherwise a fluorescent reporter was used. Percentage of virus positive cells was determined by imaging. $n = 6$ separate wells infected on the same day; error bars represent SD; ns, nonsignificant; **, $p \leq 0.01$; ***, $p \leq 0.001$; ****, $p \leq 0.0001$; ****, $p \leq 0.0001$; two-tailed $t$ test. (**J-M**) Infection of Huh-7.5 cells with viruses as indicated. siRNA knockdown of PLSCR1 vs. NTC. $n = 4$ to 16 separate wells infected on the same day; ns, nonsignificant; ****, $p \leq 0.0001$; two-tailed $t$ test. The data underlying this Figure can be found in **S1 Table**. IAV, influenza A virus; IF, immunofluorescence; NTC, nontargeting control; PLSCR1, phospholipid scramblase 1; SARS-CoV-2, Severe Acute Respiratory Syndrome Coronavirus 2.
(EPS)

**S10 Fig. Flow cytometry measuring ACE2 surface levels in live A549-ACE2 cells.**
(EPS)

**S11 Fig.** (**A**) WB against PLSCR1 and β-actin (loading control). A549-ACE2 PLSCR1 KO cells were stably transduced with FLAG-tagged Fluc, FLAG-tagged PLSCR1, or FLAG-tagged PLSCR1 H262Y. (**B**) Representative confocal images of A549-ACE2 cells treated with IFN-α2a for 48 hours. (**C**) Representative confocal images of A549-ACE2 cells infected with SARS-CoV-2 for 24 hours. KO, knockout; PLSCR1, phospholipid scramblase 1; SARS-CoV-2, Severe Acute Respiratory Syndrome Coronavirus 2; WB, western blot.
(EPS)

**S12 Fig. Representative images from the experiment shown in Fig 7C.** A549-ACE2 PLSCR1 KO cells were transduced to stably and ectopically express the indicated PLSCR1 variants. The cells were then infected for 24 hours with SARS-CoV-2 (**A**) or mock (**B**). SARS-CoV-2 N and PLSCR1 were stained and imaged. KO, knockout; PLSCR1, phospholipid scramblase 1; SARS-CoV-2, Severe Acute Respiratory Syndrome Coronavirus 2.
(EPS)

**S13 Fig. Comparison between selected SARS-CoV-2 protein interactome studies [100–109].**
(EPS)

**S14 Fig. Comparison between the relative mRNA levels of 97 hallmark IFN-α-stimulated genes and the remaining transcriptome in cell lines as indicated.** Red diamond, *PLSCR1* RNA level. For Huh-7.5 cells mRNA-seq, cells treated with 0.5 nM IFN-α2a for 24 hours as in Fig 2B. For primary human hepatocytes, cells were treated with 0.1 nM IFN-α2a for 24 hours (full data and methods to be released elsewhere). For the human tissues, data from [49]. The data underlying this Figure can be found in S1 Table.
(EPS)

**S1 Table. Data underlying Figs 1ABCEF, 2, 4ABC, 5CDEFGHIJKLM, 6ABCDEFHIJ, 7BCDE, S2ABC, S5BD, S6ABCDEFGHIJ, S7BC, S8ABC, S9ABCDEFGHIJKLM, S14.**
(XLSX)

**S2 Table. mRNA-seq on SARS-CoV-2-infected Calu-3 and Huh-7.5 cells, Normalized reads, related to Fig 1A.**
(XLSX)

**S3 Table. mRNA-seq on SARS-CoV-2-infected Calu-3 and Huh-7.5 cells, Differential Gene Expression.**
(XLSX)

**S4 Table. mRNA-seq on IFN-treated Calu-3 and Huh-7.5 cells, Normalized reads, related to Figs1B, S8C, and S13.**
(XLSX)

**S5 Table. mRNA-seq on IFN-treated Calu-3 and Huh-7.5 cells, Differential Gene Expression, related to Figs 1F and S2C.**
(XLSX)

**S6 Table. Genes expressed in 3 cell lines relevant for SARS-CoV-2 research (A549, Calu-3, and Huh-7.5 cells) and human lung cells [47–49].**
(XLSX)

**S7 Table. Arrayed CRISPR KO screen, raw data.** Channel W2 corresponds to color far red; IF staining for SARS-CoV-2 N protein.
(XLSX)

**S8 Table. Arrayed CRISPR KO screen, analyzed data.** Related to **Figs 1F** and **S2B**. Columns are as follows: *Stat.W1_SARS2_IFN_0pM*: cell growth z-score determined by nuclear staining in samples with no IFN pretreatment and with SARS-CoV-2 infection. *Stat.W1_SARS2_IFN_1pM*: cell growth z-score determined by nuclear staining in samples with IFN pretreatment and with SARS-CoV-2 infection. *Stat.W1_Uninfected_IFN_0pM*: cell growth z-score determined by nuclear staining in samples with no IFN pretreatment and no SARS-CoV-2 infection. *Stat. W2_SARS2_IFN_0pM*: SARS-CoV-2 infection z-score in samples with no IFN pretreatment and with SARS-CoV-2 infection. *Stat.W2_SARS2_IFN_1pM*: SARS-CoV-2 infection z-score in samples with IFN pretreatment and with SARS-CoV-2 infection. *Stat.W2_Uninfected_IFN_0pM*: SARS-CoV-2 infection z-score in samples with no IFNn pretreatment and no SARS-CoV-2 infection. Index: random index used for plotting. RNAseq: classification of the gene as an IFN-stimulated gene ("ISG") or not ("NS") based on the accompanying mRNA-seq data.
(XLSX)

**S9 Table. Arrayed CRISPR KO screen, summary table.**
(XLSX)

**S10 Table. GSEA on arrayed CRISPR KO screen. Related to S2D Fig.**
(XLSX)

**S11 Table. Consolidated list of human genes classified as hits in selected SARS-CoV-2 studies, full table. Related to Figs 3 and S3.**
(XLSX)

**S12 Table. Consolidated list of human genes classified as hits in selected SARS-CoV-2 studies, summary table. Related to Figs 3 and S3.**
(XLSX)

**S13 Table. Pathway analysis of hits from our arrayed screen alongside 15 unbiased pooled screens [19,31,64–67,69–71,74,75,77–80] using the STRING database [208]. Related to S4 Fig.**
(XLSX)

**S14 Table. mRNA-seq on PLSCR1 KO Huh-7.5 cells, Normalized reads, related to S7C Fig.**
(XLSX)

**S15 Table. mRNA-seq on PLSCR1 KO Huh-7.5 cells, Differential Gene Expression, related to S7C Fig.**
(XLSX)

**S16 Table. mRNA-seq on IFN-treated A549 cells, Normalized reads, related to S8B Fig.**
(XLSX)

**S17 Table. mRNA-seq on IFN-treated A549 cells, Differential Gene Expression, related to S8B Fig.**
(XLSX)

**S18 Table. Plasmids used in this study.**
(XLSX)

**S19 Table. Gene fragments used in this study.**
(XLSX)

**S20 Table. Primers used in this study.**
(XLSX)

**S1 Supporting Information. Raw images, related to S1B, S5AC, S7A, and S11 Figs.**
(ZIP)

**S2 Supporting Information. Flow cytometry gating strategy, related to S10 Fig.**
(ZIP)

**S3 Supporting Information. Flow cytometry raw data, related to S10 Fig.**
(ZIP)

## Acknowledgments

We thank Georgia McClain for reading and editing this manuscript. We thank Baptiste Milisavljevic for his assistance with the fibroblast exome-seq data. We thank the staff at the Laboratory of Virology and Infectious Disease: Ellen Castillo, Michela De Santis, Arnella Norris, Aileen O'Connell, Santa Maria Pecoraro Di Vittorio, Glen Santiago, and Sonia Shirley. Ching-Wen Chang and Lihong Liu (Columbia University, New York, USA), and Theodora Hatziioannou, Viren Baharani and Paul D. Bieniasz (The Rockefeller University, New York, USA) generously provided plasmids, instructions to generate SARS-CoV-2 spike-pseudotyped, single-cycle, replication-defective HIV-1 viruses [120,195,196], and assisted us with data interpretation. Oded Danziger and Brad R. Rosenberg (Department of Microbiology at the Icahn School of Medicine at Mount Sinai, New York, USA) kindly provided the A549-ACE2 cells [11] used in this study. FACS was conducted at the Flow Cytometry Resource Center at Rockefeller University. mRNA-seq was performed by the Genomics Resource Center at The Rockefeller University and by Novogene. Confocal microscopy was performed in the Rockefeller University's Bio-Imaging Resource Center, RRID:SCR_017791. We thank Ankit Patel, Sales Manager at Horizon Discovery.

Trim Galore was developed at The Babraham Institute by Felix Krueger, now part of Altos Labs. The Genotype-Tissue Expression (GTEx) Project [49] was supported by the Common Fund of the Office of the Director of the National Institutes of Health, and by NCI, NHGRI, NHLBI, NIDA, NIMH, and NINDS.

## Author Contributions

**Conceptualization:** Jérémie Le Pen, Gabrielle Paniccia, Eleftherios Michailidis, Brandon Razooky, Margaret R. MacDonald, Charles M. Rice.

**Data curation:** Jérémie Le Pen, Gabrielle Paniccia, Thomas S. Carroll.

**Formal analysis:** Jérémie Le Pen, Gabrielle Paniccia, Aurélie Cobat, Thomas S. Carroll.

**Funding acquisition:** Jérémie Le Pen, Gabrielle Paniccia, Eleftherios Michailidis, Margaret R. MacDonald, Charles M. Rice.

**Investigation:** Jérémie Le Pen, Gabrielle Paniccia, Volker Kinast, Marcela Moncada-Velez, Alison W. Ashbrook, Michael Bauer, H.-Heinrich Hoffmann, Ana Pinharanda, Inna Ricardo-Lax, Ansgar F. Stenzel, Edwin A. Rosado-Olivieri, Kenneth H. Dinnon, III, William C. Doyle, Catherine A. Freije, Seon-Hui Hong, Danyel Lee, Tyler Lewy, Joseph M. Luna, Avery Peace, Carltin Schmidt, William M. Schneider, Roni Winkler, Elaine Z. Yip,

John T. Poirier, Francisco J. Sànchez-Rivera, Qian Zhang, Eleftherios Michailidis, Brandon Razooky.

**Methodology:** Jérémie Le Pen, Gabrielle Paniccia, Thomas S. Carroll.

**Project administration:** Jérémie Le Pen, Charles M. Rice.

**Resources:** Chloe Larson, Timothy McGinn, Miriam-Rose Menezes, Lavoisier Ramos-Espi-ritu, Priyam Banerjee, Thomas S. Carroll, J. Fraser Glickman.

**Software:** Jérémie Le Pen, Thomas S. Carroll.

**Supervision:** Jérémie Le Pen, Jean-Laurent Casanova, J. Fraser Glickman, Margaret R. Mac-Donald, Charles M. Rice.

**Validation:** Jérémie Le Pen, Gabrielle Paniccia.

**Visualization:** Jérémie Le Pen, Gabrielle Paniccia.

**Writing – original draft:** Jérémie Le Pen.

**Writing – review & editing:** Jérémie Le Pen, Gabrielle Paniccia, Volker Kinast, Alison W. Ashbrook, Michael Bauer, Ana Pinharanda, Inna Ricardo-Lax, William M. Schneider, Margaret R. MacDonald, Charles M. Rice.

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
