## [Editor Report · Decision Letter 0]

23 Feb 2024

Dear Dr Le Pen, 

Thank you for submitting your manuscript entitled "A Genome-Wide Arrayed CRISPR Screen Reveals PLSCR1 as an Intrinsic Barrier to SARS-CoV-2 Entry" for consideration as a Research Article by PLOS Biology. 

Your manuscript has now been evaluated by the PLOS Biology editorial staff, as well as by an academic editor with relevant expertise, and I am writing to let you know that we would like to send your submission out for external peer review.

Once your full submission is complete, your paper will undergo a series of checks in preparation for peer review. After your manuscript has passed the checks it will be sent out for review. To provide the metadata for your submission, please Login to Editorial Manager (https://www.editorialmanager.com/pbiology) within two working days, i.e. by Feb 25 2024 11:59PM.

Kind regards,

Melissa

Melissa Vazquez Hernandez, Ph.D.

Associate Editor

PLOS Biology

---

## [Decision Letter · Decision Letter 1]

10 Apr 2024

Dear Dr Le Pen,

Thank you for your patience while your manuscript "A Genome-Wide Arrayed CRISPR Screen Reveals PLSCR1 as an Intrinsic Barrier to SARS-CoV-2 Entry" went through peer-review at PLOS Biology. Your manuscript has now been evaluated by the PLOS Biology editors, an Academic Editor with relevant expertise, and by three independent reviewers who would like their identity to be known; Greg Neely as R1, Hyeryun Choe as R2 and Stuart Neil as R3.

As you will see in the reports, all reviewers are positive about the relevance of the work, but still some concerns should be addressed prior to publication. Specifically, Reviewer #1 has made suggestions regarding changes in the figures and the text. Reviewer #2 requests some minor experiments to clarify some details of the relationship between PLSCR1 and TMPRSS2. Additionally, Reviewer #3 suggests additional experiments to have more mechanistic insights into PSCLR1 function.

IMPORTANT: given our anti-scooping policy and the urgency of publication, while the experimental requests from reviewer #3 will strengthen the mechanistic insights of the study, these experiments will not be required for publication.

**IMPORTANT - SUBMITTING YOUR REVISION**

*Resubmission Checklist*

*Published Peer Review*

*PLOS Data Policy*

*Blot and Gel Data Policy*

Sincerely,

Melissa

Melissa Vazquez Hernandez, Ph.D.

Associate Editor

PLOS Biology

REVIEWERS' COMMENTS

Reviewer #1: 

This manuscript describes efforts to identify new modulators of SARS-CoV-2 infection using an arrayed CRISPR KO screening approach. The screen is done using Huh-7.5 cells, which are probably not representative of a primary SARS-CoV-2 target cell, but they are naturally permissive. The investigators treat cells with IFN and then evaluate infection with authentic SARS-CoV-2. The investigators focus on one gene, PLSCR1 as an innate cellular defence component, and a similar observation was recently reported in Nature. The study is well designed and the data of reasonably high quality. 

Below are comments that are intended to improve the quality of the manuscript.

-Figure 1 might be better adapted and used instead as a visual abstract, or placed in supplemental. 

-the number of "essential genes (16790 - 16178 = 612) seems low compared to pooled CRISPR screen data. Is this because of the shorter culture time from KO to assay? 

-As I understand, the TMPRSS2-dependent fusion pathway is the primary entry pathway used in vivo. So does screening for modifiers of the endocytic pathway teach use important information about host pathogen interactions? There would be some overlap, but is this data primarily of interest to in vitro infections only? This could be discussed to help the reader understand where this data fits in the bigger picture. 

-The authors should include the total # of resistant and sensitive hits identified in the screen in the text with a reference to table s7. It would probably be easier to understand this table if z score was included here as well. For comparison between these data and previous CRISPR screens, can the authors include some information on the % of overlap with each KO? Is there significant enrichment between hit lists compared to random data? Or is each screen showing a different and complementary perspective on host virus interactions? Might need to adjust the z score for this analysis since this arrayed screen has many more hits than most of the pooled screens. It would be of general interest to know how well the various screens complement or reinforce each other. 

GSEA can often give a different perspective than just straight pathway analysis, can the authors include standard pathway analysis for the ~1000 hits at least in supplemental? This kind of analysis can help the reader understand the core mechanisms identified that themselves directly impact infection. 

-Figure 2. please add a few more sentences discussing the changes observed in 2A-C to better set up the study design. The authors should present the positive and negative control values separately so the reader can see the overall reproducibility of this approve when targeting genes with known effects. 

2E please darken the lines that set up the various response quadrants. It was slightly difficult to access these data in 2E, for some people at least its easier to just describe hits as sensitive or resistant as phenotype, vs describing sensitive as "antiviral", and resistant as "proviral", then convert the phenotype data from 2E (resistant vs sensitive) to function of the gene (proviral or antiviral) in 2F. This is not essential. Regardless, some effort to make this analysis more intuitive to the reader can increase the overall ability for more naïve readers to access the work. 

Fig 4. Are these data from a one experiment performed on one day with replicates in independent wells? Can you state in the figure legend if this is all from one experiment on one day? To say this is "independent" wouldn't it need to be done with different cells on different days? Or do you consider same cells different well infected with same virus batch "independent"?

Fig 5 probably better fits in supplemental data.

Sup Fig 5. Can you put stats on A-I? Again are these independent infections all from one round of infection done on one day? If so is this a technical replicate vs. biological replicate? Please state in the figure legend. 

Fig 7F-G. This is was a bit confusing on first pass. Can you enhance your description of this experiment in the main text? Mention 7F is analysis of A-E in text (I see its in the legend but it confused me for a second so might be easier to have a couple words in the main text). For 7G, this is plaque assay rather than just # positive cells I think. Maybe include photos of these results as well? There is room in Fig 7 and its nice to see visually what the data looks like. For all of fig 6 and 7 clarify the KO is PLSCR1 in the various panels.

Fig 8A. Label the gene PLSCR1 above the diagram to help with clarity and ease of rapid interpretation. 

Reviewer #2: 

This manuscript identifies PLSCR1 as an effective IFN-independent antiviral protein. It is clearly written, and results are well presented. A very similar paper was published several months ago (Xu et al, Nature, 2023). Compared to Xu et al., additional or somewhat different results this manuscript shows include (1) PLSCR1-mediated restriction can be overcome by TMPRSS2 over-expression, (2) more recent SARS-CoV-2 variants are less restricted by PLSCR1 than the earlier strain, and (3) use of fibroblasts derived from patients carrying heterozygous H262Y mutation in PLSCR1. To gain more insights from these differences, a couple small experiments are suggested below. Together with the study by Xu et al. and the results from the suggested experiments, this study will provide insights into a novel anti-viral mechanism. 

Major:

1. Because PLSCR1-mediated restriction can be overcome by TMPRSS2, to confirm that newer SARS-coV-2 variants are less restricted by PLSCR1 the authors should either show more recent SARS-coV-2 variant, XBB.1.5, is less dependent on TMPRSS2. If the result is different from expected, possible explanations should be given.

2. The authors stated SARS-CoV-2 infection of Huh7.5 cells does not depend on TMPRSS2, because its KO did not influence infection (lines 152-155). If Huh7.5 expresses TMPRSS2, show its expression before and after its KO. If Huh7.5 does not express TMPRSS2, it should be stated in the text.

3. As H262Y mutation is present in the region for nuclear localization signal, does this mutation affect nuclear localization of PLSCR1? If yes, the authors should discuss why restriction by PLSCR1 of beta coronaviruses whose genome does not require nuclear transport is affected by the H262Y mutation. If H262Y mutation does not affect nuclear localization of PLSCR1, describe so.

Minor:

1. Line 224: "intrinsic PLSCR1 contributes to the restriction of SARS-CoV-2, even without IFN. As we do not know if this is true in vivo, the statement should be limited to 'in vitro' or 'in cell lines'.

2. Replication-competent CHIKV and VEEV were used in the study. That they were handled only in the biosafety level 3 should be indicated.

3. Patient-derived fibroblasts carrying a heterozygous mutant (H262Y) in PLSCR1 were used in the study. The source of these cells should be given. Unless they are commercial, IRB should be in order. 

4. Are these COVID-19 patients? If not, disease name should be given. If they are COVID-19 patients, more information (the year and month cells were obtained, disease status, how long hospitalized, and etc) should be given or cite the source for such information. It may help and strengthen the authors' claim.

5. Lines 354-5 say "…evolved to bind PLSCR1 as a mechanism of immune evasion." The authors may want to elaborate on how binding to PLSCR1 helps viruses to evade immune mechanisms.

Reviewer #3: 

The manuscript by Le Pen et al presents the results of an arrayed CRISPR screen to identify host factors that are either required for, or restrict, SARS-CoV-2 replication +/- IFN treatment. Amongst other factors the authors identify is PLSCR1, a membrane protein of unknown function. PLSCR1 expression restricts viral entry and is eliminated abolished by TMPRSS2 overexpression and appears to be reduced for later SARS-CoV-2 VoCs. Lastly a coding polymorphism in PSCLR1 has been associated with SARS-CoV-2 susceptibility. The authors show that cells from individuals with this polymorphism are more susceptible to infection and ectopic expression of this mutant has an attenuated restriction actvity.

This is a well performed study and the results are of interest. The comparison with other screens is welcome. The downside is of course that a recent study has already identified that PSCLR1 is able to block the entry of SARS CoV-2. While the authors have added some further data (the human polymorph and some other viruses) the mechanistic understanding of PSCLR1 function is quite underdeveloped at present and in the light of previous publications more insight here would add value to this paper over and above a confirmatory study. The authors should address some the following:

1. I'm not particularly persuaded by the current data of the resistance of later VOCs to PSCLR1 without some more robust viral growth curves. Given that spike is sufficient to confer PSCLR1 sensitivity to HIV vectors, it should be straightforward to show that relative resistance is spike determined and if so to map the determinant in spike responsible. Since restriction is operating at entry and that TMPRSS2 expression reduces the sensitivity to PSCLR1, an obvious place to look would be the polybasic cleavage site and the adaptations associated with it. 

2. NCOA7 and IFITMs differentially affect SARS-CoV-2 entry dependent on S and proteases. Is PSCLR1 restriction associated with their activities?

3. Is there any obvious insight into why the human polymorph is less restrictive? The H262Y change maps to an apparent NLS (although whether this is real given the protein in membrane anchored is unclear). While similarly expressed, does this protein localize like the wildtype?

4. How does PSCLR1 expression affect ACE2 surface levels?

5. The specificity of PSCLR1 restriction to SARS-CoV-2 and no other enveloped virus tested including other seasonal CoVs is striking. Especially as some others appear to require it for replication. How widespread is PSCLR1 restriction amongst other sarbecovirus spikes?

Minor

* IFITM proteins are missing from figure 1.

* While "kraken" was a name that got traction on social media to describe XBB 1.5, it is part of the omicron grouping and probably shouldn't be used in scientific papers.

---

## [Editor Report · Decision Letter 2]

17 Jun 2024

Dear Dr Le Pen,

Thank you for your patience while we considered your revised manuscript "A Genome-Wide Arrayed CRISPR Screen Reveals PLSCR1 as an Intrinsic Barrier to SARS-CoV-2 Entry" for publication as a Research Article at PLOS Biology. This revised version of your manuscript has been evaluated by the PLOS Biology editors and the Academic Editor.

Based on our Academic Editor's assessment of your revision, we are likely to accept this manuscript for publication, provided you satisfactorily address the remaining points editorial points. Please also make sure to address the following data and other policy-related requests.

a) We would like to suggest the following modification to the title:

"A genome-wide arrayed CRISPR screen identifies PLSCR1 as an intrinsic barrier to SARS-CoV-2 entry that recent virus variants have evolved to resist"

Please supply the numerical values either in the a supplementary file or as a permanent DOI’d deposition for the following figures:

Figure 1ABCEF, 2, 4ABC, 5CDEFGHIJKLM, 6ABCDEFHIJ, 7BCDE, S2ABC, S5BD, S6ABCDEFGHIJ, S7BC, S8ABC, S9ABCDEFGHIJKLM, S14

c) Please cite the location of the data clearly in all relevant main and supplementary Figure legends, e.g. “The data underlying this Figure can be found in S1 Data” or “The data underlying this Figure can be found in https://doi.org/10.5281/zenodo.XXXXX”

d) We require the original, uncropped and minimally adjusted images supporting all blot and gel results reported in the Figures:

Figure S1B, S5AC, S7A, S11

We will require these files before a manuscript can be accepted so please prepare and upload them now. Please carefully read our guidelines for how to prepare and upload this data: https://journals.plos.org/plosbiology/s/figures#loc-blot-and-gel-reporting-requirements

e) “For figures containing FACS data (Figure S10), please provide the FCS files and a picture showing the successive plots and gates that were applied to the FCS files to generate the figure. We ask that you please deposit this data in the FlowRepository (https://flowrepository.org/) and provide the accession number/URL of the deposition in the Data Availability Statement in the online submission form.”

e) Please ensure that your Data Statement in the submission system accurately describes where your data can be found and is in final format, as it will be published as written there.

f) We thank you for uploading the supplemental data to Dryad. However, we were not able to access it and evaluate the data deposited there. Please make sure to allow access before submitting it again.

f) Per journal policy, if you have generated any custom code during the curse of this investigation, please make it available without restrictions upon publication. Please ensure that the code is sufficiently well documented and reusable, and that your Data Statement in the Editorial Manager submission system accurately describes where your code can be found.

We expect to receive your revised manuscript within two weeks. 

*Published Peer Review History*

*Press*

Sincerely,

Melissa

Melissa Vazquez Hernandez, Ph.D.

Associate Editor

PLOS Biology

---

## [Editor Report · Decision Letter 3]

25 Jul 2024

Dear Jérémy,

I hope you, your wife and the baby are doing great. Thank you for the submission of your revised Research Article "A genome-wide arrayed CRISPR screen identifies PLSCR1 as an intrinsic barrier to SARS-CoV-2 entry that recent virus variants have evolved to resist" for publication in PLOS Biology. On behalf of my colleagues and the Academic Editor, Ken Cadwell, I am pleased to say that we can in principle accept your manuscript for publication, provided you address any remaining formatting and reporting issues. These will be detailed in an email you should receive within 2-3 business days from our colleagues in the journal operations team; no action is required from you until then. Please note that we will not be able to formally accept your manuscript and schedule it for publication until you have completed any requested changes.

IMPORTANT: Thank you for attending to the previous editorial requests and for providing all underlying data and supplementary tables. However, the tables were uploaded as text files. For accesability to the users, please provide all supplementary tables as Excel files, or at least .csv files, and re-upload them. My colleagues in the operations team will mention this when they contact you in the next days. Final acceptance for publication will not be possible without it. 

PRESS

Sincerely, 

Melissa

Melissa Vazquez Hernandez, Ph.D., Ph.D.

Associate Editor

PLOS Biology
